# Measuring the Reliability of Causal Probing Methods: Tradeoffs, Limitations, and the Plight of Nullifying Interventions

## Abstract

Causal probing aims to analyze large language models (or other foundation models) by examining how modifying their representation of various latent properties using interventions derived from probing classifiers impacts their outputs. Recent works have cast doubt on the theoretical basis of several leading causal probing intervention methods, but it has been unclear how to systematically evaluate the effectiveness of probing interventions in practice. To address this, we formally define and quantify two key causal probing desiderata: *completeness* (how thoroughly the representation of the target property has been transformed) and *selectivity* (how little other properties have been impacted). We introduce an empirical analysis framework to measure and evaluate completeness and selectivity, allowing us to make the first direct comparisons of the reliability of different families of causal probing methods (e.g., linear vs. nonlinear or counterfactual vs. nullifying interventions). Our experimental analysis shows that: (1) there is an inherent tradeoff between completeness and selectivity, (2) no leading probing method is able to consistently satisfy both criteria at once, and (3) across the board, nullifying interventions are far less complete than counterfactual interventions, which suggests that nullifying methods may not be an effective approach to causal probing.

## 1 Introduction

What latent properties do large language models (LLMs) learn to represent, and how are these representations leveraged by these models? Causal probing aims to answer this question by first transforming a model's embedding representations of some latent property of interest (e.g., parts-of-speech) by intervening on probes trained to predict the property, feeding embeddings back into the LLM, and evaluating how the model's behavior on downstream tasks changes in the context of these interventions (Geiger et al., 2020; Ravfogel et al., 2020; Elazar et al., 2021; Tucker et al., 2021; Lasri et al., 2022; Davies et al., 2023; Zou et al., 2023). This approach assumes that such interventions will affect the behavior of models that leverage the target property to perform the task, and that a negligible impact indicates that models are not using the property. But such conclusions should only be drawn if we can be confident that the intervention method has fully and precisely carried out the intended transformation (Davies & Khakzar, 2024). Indeed, prior works have raised serious doubts about the utility of causal probing, suggesting that interventions may have a greater impact on the representation of non-targeted properties than on that of the target property (Kumar et al., 2022), and that the original value of the property may still be recoverable from intervened embeddings (Elazar et al., 2021; Ravfogel et al., 2022b). However, it is unclear whether observations about the shortcomings of one intervention generalize to other types of interventions, since there is currently no generally accepted approach for how to empirically compare the reliability of different intervention methodologies.

Thus, our main goal in this work is to work toward a systematic understanding of the effectiveness and limitations of current causal probing methodologies. Specifically, we propose an empirical analysis framework to evaluate the *reliability* of causal probing according to two key desiderata:

1. *Completeness*: interventions should fully remove the representation of targeted properties.

2. *Selectivity*: interventions should not impact non-targeted properties.

We measure the extent to which different types of interventions are complete and selective by using "oracle probes" that allow us to (approximately) estimate the impact of an intervention on both targeted and non-targeted properties. We apply our framework to several leading causal probing intervention methodologies, finding that they each show a clear tradeoff between these criteria, and that no method is able to satisfy them both simultaneously. We also show that, when intervening on a property that is necessary to solve a specific task, the most complete and reliable interventions lead to the most consistent and substantial changes in LLM task performance. Finally, we observe that, across all methods we study, counterfactual causal probing interventions are universally more complete and reliable than nullifying interventions. While this result is consistent with the findings of earlier work criticizing nullifying interventions (see Section 2), it has not previously been possible to directly compare them with counterfactual interventions, and evidence against nullifying interventions has often been used to argue against causal probing more broadly. However, our results, as enabled by our empirical evaluation framework, show that the most serious limitations are generally only present for nullifying methods, and that counterfactual methods constitute a more reliable approach to causal probing.

## 2 BACKGROUND AND RELATED WORK

**Structural Probing** The goal of *structural probing* (Hewitt & Manning, 2019; Maudslay et al., 2020; Belinkov et al., 2020) is to analyze which properties (e.g., part-of-speech, sentiment labels, etc.) are represented by a deep learning model (e.g., LLM) by training probing classifiers to predict these properties from latent embedding representations (Belinkov, 2022). Given, say, an LLM $M$, input token sequence $\mathbf{x} = (x_1, ..., x_N)$, and embeddings $\mathbf{h}^l = M_l(\mathbf{x})$ of input $\mathbf{x}$ at layer $l$ of $M$, suppose $Z$ is a latent property of interest that takes a discrete value $Z = z$ for input $\mathbf{x}$. The goal of structural probing is to train a classifier $g_Z: M_l(\mathbf{x}) \mapsto z$ to predict the value of $Z$ from $\mathbf{h}^l$. On the most straightforward interpretation, if $g_Z$ achieves high accuracy on the probe task, then the model is said to be "representing" $Z$ (Belinkov, 2022). An important criticism of such methodologies is that *correlation does not imply causation* – i.e., that simply because a given property can be predicted from embedding representations does not mean that the model is using the property in any way (Hewitt & Liang, 2019; Elazar et al., 2021; Belinkov, 2022; Davies et al., 2023).

**Causal Probing** A prominent response to this concern has been *causal probing*, which uses structural probes to remove or alter that property in the model's representation, and measuring the impact of such interventions on the model's predictions (Elazar et al., 2021; Tucker et al., 2021; Lasri et al., 2022; Davies et al., 2023). Specifically, causal probing performs interventions $\mathrm{do}(Z)$ that modify $M$'s representation of $Z$ in the embeddings $\mathbf{h}^l$, producing $\hat{\mathbf{h}}^l$, where interventions can either encode a counterfactual value $Z = z'$ (denoted $\mathrm{do}(Z = z')$ where $z \neq z'$), or remove the representation of $Z$ entirely (denoted $\mathrm{do}(Z = 0)$). Following the intervention, modified embeddings $\hat{\mathbf{h}}^l$ are fed back into $M$ beginning at layer $l + 1$ to complete the forward pass, yielding intervened predictions $P_M(\cdot \mid \mathbf{x}, \mathrm{do}(Z))$. Comparison with the original predictions $P_M(\cdot \mid \mathbf{x})$ allows one to measure the extent to which $M$ uses its representation of $Z$ in computing $P_M(\cdot \mid \mathbf{x})$.

**Causal Probing: Limitations** However, prior works have indicated that information about the target property that should have been completely removed may still be recoverable by the model (Elazar et al., 2021; Ravfogel et al., 2022b; 2023), in which case interventions are not complete; or that most of the impact of interventions may actually be the result of collateral damage to correlated, non-targeted properties (Kumar et al., 2022), in which case interventions are not selective. How seriously should we take such critiques? We observe several important shortcomings in each of these prior studies on the limitations of causal probing interventions:

1. These limitations have only been empirically demonstrated for the task of removing information about a target property from embeddings such that the model *cannot be fine-tuned to use the property for downstream tasks* (Kumar et al., 2022; Ravfogel et al., 2022b; 2023). But considering that the goal of causal probing is to interpret the behavior of an existing pre-trained model, the question is not whether models *can* be fine-tuned to use the property;

it is whether models *already* use the property without task-specific fine-tuning, which has not been addressed in prior work. Do we observe the same limitations in this context?

2. These limitations have only been studied in the context of linear nullifying interventions (e.g., Ravfogel et al. 2020; 2022a), despite the proliferation of other causal probing methodologies that have been developed since, including nonlinear (Tucker et al., 2021; Ravfogel et al., 2022b; Shao et al., 2022; Davies et al., 2023) and counterfactual interventions (Ravfogel et al., 2021; Tucker et al., 2021; Davies et al., 2023) (see Section 4.4). Do we observe the same limitations for, e.g., nonlinear counterfactual interventions?

In this work, we answer both questions by providing a precise, quantifiable, and sufficiently general definition of completeness and selectivity that it is applicable to *all* such causal probing interventions, and carry out extensive experiments to evaluate representative methods from each category of interventions when applied to a pre-trained LLM as it performs a zero-shot prompt task.

**Causal Probing: Evaluation** While we are the first to define and measure the completeness and selectivity of causal probing interventions, a few other works have also aimed to evaluate causal probing approaches (or closely related methods). Closest to our work, Huang et al. (2024) evaluates interchange interventions over *factual associations* – e.g., modifying a model's representation to encode located_in(France, Asia) rather than located_in(France, Europe), where interchange interventions extract the representation of association $A_1$ for entity $E_1$, then swap out the representation of association $A_2$ with entity $E_2$ to encode $A_1$ instead (Meng et al., 2022). Such interventions operate at the level of individual entities (e.g., France, Asia, or Europe) rather than general latent properties (e.g., part-of-speech) which have categorical values that are each taken by many different inputs, as studied in causal probing.

## 3 EVALUATING CAUSAL PROBING RELIABILITY

Recall that our main goal in this work is to evaluate intervention reliability in terms of completeness (completely transforming $M$'s representation of some target property $Z_i$) and selectivity (minimally impacting $M$'s representation of other properties $Z_j \neq Z_i$).[1] Given that we cannot directly inspect what value $M$ encodes for any given property $Z_i$, how can we determine the extent to which interventions have fulfilled either criterion? In order to formally define these criteria, we introduce the notion of *oracle probes*.

**Oracle Probes** We define an oracle probe $o$ as a structural probe that returns a distribution $P_o(Z \mid \mathbf{h}^l)$ over the values of property $Z$, and we interpret $P_o(Z = z \mid \mathbf{h}^l)$ as the degree to which the model's representations $\mathbf{h}^l = M_l(\mathbf{x})$ encodes a belief that input $\mathbf{x}$ has the property $Z = z$. So, if $\mathbf{h}^l$ encodes value $Z = \hat{z}$ with complete certainty, $o$ should return a degenerate distribution $P_o(Z \mid \mathbf{h}^l) = \mathbb{1}(Z = \hat{z})$, whereas we would expect a uniform distribution $P_o(Z \mid \mathbf{h}^l) = \mathcal{U}(Z)$ if $\mathbf{h}^l$ does not encode property $Z$ at all.[2] Naturally, a perfect oracle does not exist in practice, so any practical implementation must approximate it (see Section 4.3). However, a sufficiently high-quality oracle probe enables us to measure in practice how well various intervention methodologies perform the desired intervention.[3]

**Completeness** If a counterfactual intervention $\mathrm{do}(Z = z')$ is perfectly *complete*, then it should produce $\mathbf{h}^{l^*}_{Z=z'}$ that fully transforms $\mathbf{h}^l$ from encoding value $Z = z$ to encoding counterfactual value $Z = z' \neq z$. Thus, after performing the intervention, oracle $o$ should emit $P_o(Z = z' \mid \mathbf{h}^{l^*}_{Z=z'}) =$

---

[1] Note that, throughout this paper, we use selectivity in the sense described by Elazar et al. (2021), and not other probing work such as Hewitt & Liang (2019), where it instead refers to the gap in performance between probes trained to predict real properties versus nonsense properties.

[2] An oracle probe's prediction is subtly different from the prediction an arbitrary classifier should make in the absence of any evidence about $Z$: such a classifier should revert to the empirical distribution $\hat{P}(Z)$.

[3] Note that (approximated) oracle probes are never the same probes as those used to compute interventions. In our experimental formulation, we use completely disjoint training data and multiple different probe architectures to ensure that oracle probes can serve as effective neutral arbiters of intervention reliability. See Section 4.3 for further discussion.

$P_Z^*(Z = z') = 1$. For nullifying interventions $\mathrm{do}(Z = 0)$, a perfectly complete representation $\mathbf{h}_{Z=0}^{l*}$ should not encode $Z$ at all: $P_o(Z \mid \mathbf{h}_{Z=0}^{l*}) = P_Z^* = \mathcal{U}(Z)$.[4]

We can use any distributional distance metric $\delta(\cdot, \cdot)$ bounded by [0, 1] to determine how far the observed distribution $\hat{P}_Z = P_o(Z \mid \hat{\mathbf{h}}_Z^l)$ is from the "goal" distribution $P_Z^*$. Throughout this work, we use total variation (TV) distance, but any other such metric can be substituted in its place. Importantly, defining completeness $c(\hat{\mathbf{h}}_Z^l)$ in this way allows us to directly compare counterfactual and nullifying distributions: in both cases, $0 \leq c(\hat{\mathbf{h}}_Z^l) \leq 1$, where attaining 1 means the intervention had its intended effect in transforming the encoding of $Z$. Finally, for a given set of test embeddings $\mathbf{H}^l = \{\mathbf{h}^{l,k}\}_{k=1}^n$, the aggregate completeness over this test set $C(\mathbf{H}_Z^l)$ is the average completeness $c(\hat{\mathbf{h}}_Z^{l,i})$ across all $\mathbf{h}^{l,k} \in \mathbf{H}^l$.

For **counterfactual interventions**, we measure completeness as:

$$c(\hat{\mathbf{h}}_Z^l) = 1 - \delta(\hat{P}, P_Z^*) \tag{1}$$

If the intervention is perfectly complete, then $\hat{P} = P_Z^*$ and $c(\hat{\mathbf{h}}_Z^l) = 1$. On the other hand, if $\hat{P}$ is maximally different from the goal distribution $P_Z^*$ (e.g., $\hat{P} = P_o(Z = z \mid \hat{\mathbf{h}}_{Z=z'}^l) = 1$), then $c(\hat{\mathbf{h}}_Z^l) = 0$. For properties with more than two possible values, completeness is computed by averaging over each possible counterfactual value $z_1', ..., z_k' \neq z$, yielding $c(\hat{\mathbf{h}}_Z^l) = \frac{1}{k}\sum_{i=1}^k \hat{c}(\mathbf{h}_{Z=z_i'}^l)$.

For **nullifying interventions**, we measure completeness as:

$$c(\hat{\mathbf{h}}_Z^l) = 1 - \frac{k}{k-1} \cdot \delta(\hat{P}, P_Z^*) \tag{2}$$

where $k$ is the number of values $Z$ can take. The normalizing factor is needed because $P_Z^*$ is the uniform distribution over $k$ values and hence $0 \leq \delta(\hat{P}, P_Z^*) \leq 1 - \frac{1}{k}$.

**Selectivity**   If an intervention on property $Z_i$ is *selective*, the intervention should not impact $M$'s representation of any non-targeted property $Z_j \neq Z_i$. Thus, for both counterfactual and nullifying interventions, oracle $o$'s prediction for any such $Z_j$ should not change after the intervention.

To measure the selectivity of a modified representation $\hat{\mathbf{h}}_{Z_i}^l$ with respect to $Z_j$, denoted $s_j(\hat{\mathbf{h}}_{Z_i}^l)$, we can again measure the distance between the observed distribution $\hat{P} = P_o(Z_j \mid \hat{\mathbf{h}}_{Z_i}^l)$ and the original (non-intervened) distribution $P = P_o(Z_j \mid \mathbf{h}^l)$:

$$s_j(\hat{\mathbf{h}}_{Z_i}^l) = 1 - \frac{1}{m} \cdot \delta(\hat{P}, P) \quad \text{where}$$
$$m = \max\left(1 - \min(P), \max(P)\right) \tag{3}$$

Since $0 \leq \delta(\hat{P}, P) \leq m$, we divide by $m$ to normalize selectivity to $0 \leq s_j(\hat{\mathbf{h}}_{Z_i}^l) \leq 1$. If multiple non-targeted properties $Z_{j_1}, ..., Z_{j_{\max}}$ are being considered, selectivity $s(\hat{\mathbf{h}}_{Z_i}^l)$ is computed as the average over all such properties $s_{j_m}(\hat{\mathbf{h}}_{Z_i}^l)$. Finally, analogous to completeness, the aggregate selectivity over a set of test embeddings $\mathbf{H}^l = \{\mathbf{h}^{l,k}\}_{k=1}^n$, denoted $S(\mathbf{H}_{Z_i}^l)$, is the average selectivity $s(\hat{\mathbf{h}}_{Z_i}^{l,k})$ across all $\mathbf{h}^{l,k} \in \mathbf{H}^l$.

**Reliability**   Since completeness and selectivity can be seen as a trade-off, we define the overall reliability of an intervention $R(\hat{\mathbf{H}}^l)$ as the harmonic mean of $C(\hat{\mathbf{H}}^l)$ and $S(\hat{\mathbf{H}}^l)$:

$$R(\hat{\mathbf{H}}_Z^l) = 2\frac{C(\hat{\mathbf{H}}_Z^l) \cdot S(\hat{\mathbf{H}}_Z^l)}{C(\hat{\mathbf{H}}_Z^l) + S(\hat{\mathbf{H}}_Z^l)} \tag{4}$$

---

[4]Note that this is only expected when using nullifying interventions for *causal probing* – i.e., when intervening on a model's representation and feeding it directly back into the model to observe how the intervention modifies its behavior. When considering nullifying interventions for *concept removal* (which is a more common setting), a more appropriate "goal" distribution $P_Z^*$ would be $P(Z)$, the label distribution. See Appendix B for further discussion.

This is analogous to the F1-score as the harmonic mean of precision and recall: since a degenerate classifier can achieve perfect recall and low precision by always predicting the positive class, the F1-score heavily penalizes models that trade off recall for precision (or vice versa). A degenerate intervention can achieve perfect selectivity and low completeness by performing no intervention at all, which should similarly be heavily penalized, as we do here for reliability score $R(\hat{\mathbf{H}}^l)$.

# 4 EXPERIMENTAL SETTING

## 4.1 LANGUAGE MODEL: BERT

We test our framework by carrying out an extensive range of experiments to analyze current causal probing interventions in the context of BERT (Devlin et al., 2019). We opt for BERT because it is very well-studied in the context of language model interpretability (Rogers et al., 2021), particularly in causal probing (Ravfogel et al., 2020; Elazar et al., 2021; Ravfogel et al., 2021; Lasri et al., 2022; Ravfogel et al., 2022b; 2023; Davies et al., 2023). We further motivate this decision in Appendix A.

## 4.2 TASK: SUBJECT-VERB AGREEMENT

To test our evaluation framework experimentally, we need a task with a clear causal structure. We start with the simplest possible case, where labels are fully determined by the value of a single binary causal variable $Z_c$, with a second binary environmental variable $Z_e$ that influences inputs and may be spuriously correlated with the label (meaning that LLMs might still leverage it in performing the task). By intervening on each of these variables, we can calculate how well the target variable is damaged (completeness) and how little the opposite variable is damaged (selectivity). We also want a task that BERT performs well, allowing us to assess the extent to which its performance is attributable to its representation of the causal variable $Z_c$ versus the spuriously-correlated environmental variable $Z_e$.

To fulfill both criteria, we select the cloze prompting task of **subject-verb agreement**, which has also been the subject of analysis in multiple prior causal probing works (Lasri et al., 2022; Ravfogel et al., 2021). Each instance in this task takes the form $\langle \mathbf{x}_i, y_i \rangle$ where $\mathbf{x}_i$ is a sentence such as "the girl with the keys [MASK] the door," and the task of the LLM is to predict $P_M(y_i \mid \mathbf{x}) > P_M(y_i' \mid \mathbf{x})$ (here, that $y$ = "locks" rather than "lock"). The causal variable $Z_c$ for this task is the number of the subject, because (grammatically) this is the only variable the number of the verb depends on. The values $Z_c$ can take are Sg and Pl. The environmental variable $Z_e$ is the number of the noun immediately preceding the [MASK] token when that noun is not the subject. Most commonly, this is the object of a prepositional phrase (e.g., in the earlier example, this would be "keys" in the phrase "with the keys"). The values $Z_e$ can take are also Sg and Pl.

We use the LGD subject-verb agreement dataset (Linzen et al., 2016), which consists of $> 1$M naturalistic English sentences from Wikipedia. We use syntax annotations to extract values for the environmental variable: if the part-of-speech of the word immediately preceding the [MASK] token is a noun, and it is the object of a preposition (i.e., not the subject), then its number defines $Z_e$. About 83% of the sentences do not have a prepositional object preceding [MASK], and so are only relevant for causal interventions. Because subject-verb agreement is a prediction task over two verb forms, and as BERT is a masked-language model, we only use sentences for which both forms of the target verb are in BERT's vocabulary. We use 40% of the examples to train the oracle probe (of which 5% is used for validation data), 40% to train the interventions (again, 5% is used for validation data), and 20% as a test set. (The contingency table for the test set is shown in Table 2.)

Finally, for all experiments, we analyze [MASK] embeddings $\mathbf{h}^l_{\texttt{[MASK]}}$ from BERT's final layer $l = |L|$ immediately before it is fed into the masked language modeling head. We do this because, for earlier layers, any information about the target property $Z$ removed or modified by interventions over the [MASK] embedding may be recoverable from embeddings of other tokens (see Elazar et al., 2021), as attention blocks allow the model to pool information from the contextualized embeddings of other tokens.

### 4.3 ORACLE PROBE APPROXIMATION

**Oracle Probe Architecture**    Before we can measure how well causal probing interventions satisfy completeness and selectivity, we must first provide a suitable approximation of oracle probe $o$. What probe would serve as the best approximation? As our goal with oracle probes is to measure the impact of causal probing interventions on the representation of the property in LLM embeddings (and not to measure how well models encode a property of interest to begin with, as in structural probing), we follow Pimentel et al. (2020)'s argument that more expressive, higher-performing probes are a better choice for measuring the mutual information between embedding representations and properties of interest, and opt to use more expressive probes $\hat{o}$ to approximate the oracle $o$. Specifically, we implement $\hat{o}$ as a multi-layer perceptron (MLP), and select hyperparameters that yield the probe with the highest validation-set accuracy using grid search (see Appendix C.2 for further details).

**Oracle Probe Training**    Oracle probes are trained on 40% of the available LGD data, which is entirely separate from the data used to train interventions. As these probes are used to assess the representation of $Z_c$ and $Z_e$ in latent vector representations, it is important that their training data does not overlap with the training data of probes used for interventions. Finally, we require oracle probes to be trained on data where there is 0 correlation between $Z_c$ and $Z_e$ by selecting the largest possible subset of the oracle probe training data such that (1) there is no correlation between $Z_c$ and $Z_e$, and (2) the label distribution of the target property is preserved. This is crucial, because any spurious correlation between these variables could lead to a probe that is trained on property $Z_c$ to also partially rely on representations of $Z_e$ (Kumar et al., 2022).

### 4.4 INTERVENTIONS

In this section we describe each intervention that we study. In all cases, probes used for interventions are trained on a fully disjoint train set from that used to train the oracle probes.

**Nullifying Interventions**    We experiment with two nullifying interventions:

1. INLP (Ravfogel et al., 2020) removes all linearly-predictive information about the target property from embeddings by iteratively training linear probes to predict the property, projecting embeddings into the nullspace of the classifier, training a new classifier on the projected embeddings, etc. until they are no longer linearly predictive of the target property.
2. RLACE (Ravfogel et al., 2022a) identifies a minimal-rank bias subspace to remove linearly-predictive information by solving a constrained, linear minimax game, in which a probe aims to predict the concept while an adversary hinders it by projecting input embeddings into a lower-dimensional subspace.

Additional details regarding our implementation of each nullifying intervention are in Appendix C.3.

**Counterfactual Interventions**    We experiment with four counterfactual interventions: three gradient-based interventions (GBIs) (Davies et al., 2023),[5] and AlterRep. GBIs train traditional structural probes over embeddings and perform interventions by attacking these probes using gradient-based adversarial attacks, which perturb the embeddings to minimize the loss of the structural probe with respect to the target counterfactual value $Z = z'$ within an $L_\infty$-ball with radius $\epsilon$ around the original embedding. We explore three popular adversarial attacks:

1. FGSM (Goodfellow et al., 2015) takes a single gradient step of magnitude $\epsilon$ in the direction that minimizes the probe loss with respect to the target class $z'$.
2. PGD (Madry et al., 2017) iteratively minimizes the probe loss with respect to $z'$ by performing gradient descent within a $L_\infty$-ball of radius $\epsilon$, projecting back onto its surface if a given step takes the embedding outside the ball.
3. AutoAttack (Croce & Hein, 2020) minimizes the probe loss with respect to $z'$ by ensembling over several adversarial attacks: two black-box attacks (FAB and Square Attack) and APGD (a version of PGD utilizing momentum in gradient steps).

---

[5]Note that Tucker et al. (2021) also define a similar methodology without explicit use of adversarial attacks or $\epsilon$-constraints. As Davies et al. (2023) provide a more general formulation explicitly in the language of adversarial attacks (including $\epsilon$-constraints, etc.), we follow their formulation.

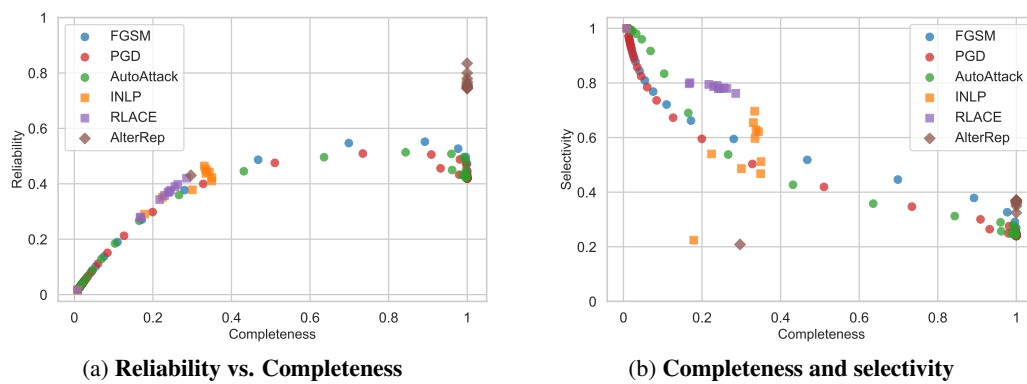

(a) **Reliability vs. Completeness**  (b) **Completeness and selectivity**

Figure 1: **All interventions.** Reliability, selectivity, and completeness tradeoffs for all interventions. Each point corresponds to a different hyperparameter setting ($\epsilon$ for GBIs, rank $r$ for INLP and RLACE, and $\alpha$ for AlterRep). Dot shape indicates intervention type (circle for non-linear counterfactual, square for linear nullifying, and diamond for linear counterfactual).

AlterRep (Ravfogel et al., 2021) begins by performing INLP, then uses the sequence of linear classifiers to project embeddings into the classifiers' rowspaces and adds the rowspace projection to the nullified representations, placing embeddings on the target (counterfactual) side of the classifiers' separating hyperplanes. (See Appendix C.3 for additional details regarding our implementation of all 3 GBIs and AlterRep.)

### 4.5 MEASURING RELIABILITY & IMPACT ON MODEL BEHAVIOR

**Measuring Reliability**  After intervening on $Z_c$ to obtain representations $\hat{\mathbf{h}}_{Z_c}^{|L|}$, we evaluate how well $Z_c$ was removed (completeness) and how little it damaged $Z_e$ (selectivity) using the approximated oracle probes $\hat{o}$ to compute $P_{\hat{o}}(Z \mid \hat{\mathbf{h}}_{Z_c}^{|L|})$ for $Z \in \{Z_c, Z_e\}$. Scores for completeness (Equations 1 and 2), selectivity (Equation 3), and reliability (Equation 4) are reported as the average across all examples in the test set.

**Impact on Model Behavior**  The ultimate goal of causal probing is to measure a model $M$'s use of a property $Z$ by comparing intervened predictions $P_M(\cdot \mid \mathbf{x}, \mathrm{do}(Z))$ to its original predictions $P_M(\cdot \mid \mathbf{x})$. The experimental framework described to this point aims to measure the reliability of the interventions themselves, a prerequisite to making claims about the underlying model. Thus, while the main focus of our work is to *evaluate the reliability* of interventions as a prerequisite for more rigorous causal probing – and *not* to perform interventions on a given model in order to make any specific claims about the model itself – it is nonetheless important to consider how the completeness, selectivity, and reliability of a given intervention relate to the impact on model behavior observed using the intervention.

For each intervention, we feed the final-layer intervened vectors $\hat{\mathbf{h}}_{Z_c}^{|L|}$ into BERT's prediction head, and predict the singular or plural form of the missing verb based on which has higher probability. We compute accuracy across all test examples, and compare this "intervened" accuracy with the original task accuracy for each intervention, a common protocol for causal probing (Elazar et al., 2021; Lasri et al., 2022; Davies et al., 2023). Thus, instead of measuring the impact of interventions on oracle probes, we consider their impact on the language model itself predicting the correct output.

## 5 EXPERIMENTAL RESULTS

We first verify that our oracle MLP probes are able to pick up on the properties of interest: the oracle probe for causal variable $Z_c$ achieves 99.4% accuracy, and the probe for $Z_e$ attains 88.4% accuracy.

|  | $C(\hat{\mathbf{H}}_Z^l)$ | $S(\hat{\mathbf{H}}_Z^l)$ | $R(\hat{\mathbf{H}}_Z^l)$ | $x_{opt}$ |
|---|---|---|---|---|
| FGSM | 0.8923 | 0.3994 | 0.5518 | $\epsilon = 0.112$ |
| PGD | 0.7343 | 0.3897 | 0.5092 | $\epsilon = 0.112$ |
| AutoAttack | 0.8433 | 0.3692 | 0.5136 | $\epsilon = 0.112$ |
| AlterRep | 1.0000 | 0.7842 | **0.8346** | $\alpha = 0.1$ |
| INLP | 0.3308 | 0.7792 | 0.4644 | $r = 8$ |
| RLACE | 0.2961 | 0.7782 | 0.4290 | $r = 33$ |

Table 1: **All causal interventions.** Completeness, selectivity, and reliability scores for each intervention we consider, when the intervention is performed against causal variable $Z_c$. Optimal hyperparameter $x_{opt}$ is chosen at the value that maximizes reliability for each respective method. Counterfactual methods are grouped above the double line; nullifying methods are below it.

## 5.1 COMPLETENESS, SELECTIVITY, & RELIABILITY

Each intervention has a respective hyperparameter ($\epsilon$ for GBIs, rank $r$ for INLP and RLACE, and $\alpha$ for AlterRep), where increasing the value of the hyperparameter corresponds to increased damage to representations. Thus, each hyperparameter setting yields a different value of completeness, selectivity, and reliability for a given intervention method. Figure 1a plots reliability against completeness for all methods and hyperparameter settings, showing that once a certain amount of damage has been done to the target property (by raising the value of the hyperparameter), reliability flattens. This is due to excessive collateral damage done by the intervention, resulting in lower selectivity. This can be seen in Figure 1b, which plots selectivity against completeness for each method. There is a clear general tradeoff between the two desiderata: an excessively complete intervention may come at the cost of doing too much damage to other variables (i.e., low selectivity).

Table 1 shows these metrics for each intervention under the hyperparameter that yields the highest reliability for each intervention performed against $Z_c$ (results at each hyperparameter setting for each method are available in Appendix D.2). AlterRep (the only intervention we consider that is both linear and counterfactual) achieves the highest reliability score, with perfect completeness and reasonably high selectivity. The nonlinear counterfactual methods (FGSM, PGD, and AutoAttack) all perform very similarly to each other, but are significantly less reliable than AlterRep. Finally, the linear nullifying methods INLP and RLACE are least reliable, largely due to their low completeness.

## 5.2 TASK ACCURACY

Figures 2a and 2b show the change in BERT's task accuracy against completeness and reliability for each intervention, where each point corresponds to a different hyperparameter setting. Change in task accuracy is calculated as the difference between the original task accuracy (98.62%) and the "intervened" task accuracy (the accuracy achieved when feeding the intervened vector into BERT's prediction layer).

For most methods and hyperparameter values, both figures depict a generally increasing trend: as intervention completeness and reliability increases, task accuracy is usually increasingly affected. The exceptions to this pattern are as follows: first, AlterRep shows a very substantial impact on task accuracy (almost 98.62, corresponding to a drop to $0\%$ accuracy) for most hyperparameter values, consistent with its being the most reliable method. Second, the points at which the GBIs (FGSM, PGD, and AutoAttack) achieve the highest change in task accuracy are *not* at the highest reliability values for these methods, resulting in a backward curve (increasing task impact while reliability decreases) visible at the top of Figure 2b. Further investigation (see Appendix D.2) shows that these points correspond to hyperparamter $\epsilon$ being raised past the point of maximum reliability: the target property is severely damaged (resulting in high change in task accuracy), but significant collateral damage has also been done (i.e., there is low selectivity, leading to the decrease in overall reliability). Finally, RLACE shows a similar impact on task accuracy even for different completeness and reliability scores, which is the result of its comparatively "noisy" equilibrium in reliability and completeness for high rank $r$ (see Appendix D.2).

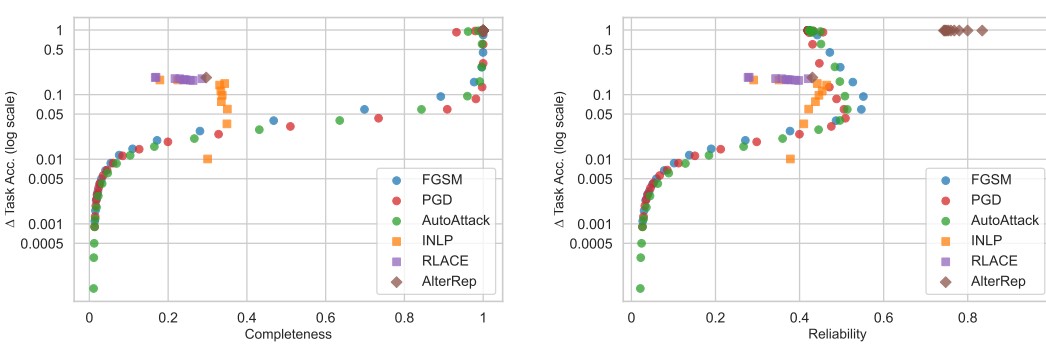

(a) **Change in task accuracy vs. completeness**  (b) **Change in task accuracy vs. reliability**

Figure 2: **Change in task accuracy.** Difference between subject-verb agreement task accuracy with original embeddings and task accuracy with intervened embeddings, where interventions are performed against $Z_c$. Each point corresponds to a different hyperparameter setting ($\epsilon$ for GBIs, rank $r$ for INLP and RLACE, and $\alpha$ for AlterRep). Dot shape indicates intervention type (circle for non-linear counterfactual, square for linear nullifying, and diamond for linear counterfactual).

## 6 DISCUSSION

**Tradeoff: Completeness vs. Selectivity**  Figure 1b shows that counterfactual methods are able to achieve near perfect completeness, for both linear (AlterRep) and nonlinear (GBI) interventions; but AlterRep has much higher selectivity while maintaining perfect completeness. These findings suggest that linear methods may indeed be preferred for causal probing because their limited expressivity prevents them from memorizing spurious associations in probe training data (Hewitt & Liang, 2019), potentially making them less likely to damage spurious (environmental) features, meaning that they would be more selective. Another possibility is that linear methods can only achieve such high completeness in these experiments because we are considering embeddings from the final layer, which have been argued to exhibit a greater degree of linearity (Alain & Bengio, 2017). However, despite the impressive results of AlterRep, its reliability score still peaks at $0.8346$ for the optimal value of hyperparamteter $\alpha$, indicating that there is still room for improvement. In particular, while GBIs can be precisely calibrated to manage the completeness/selectivity tradeoff and can reach full completeness for high $\epsilon$, they never achieve an overall reliability above $0.5518$, meaning that improving selectivity for nonlinear interventions is very much an open research question.

In contrast to counterfactual methods, nullifying interventions never achieve high completeness (peaking at $0.3308$ for INLP at the high watermark for reliability, $0.4644$), but tend to have much higher selectivity (at least relative to GBIs). This is likely because both INLP and RLACE are explicitly optimized to minimize collateral damage while removing the target representation (Ravfogel et al., 2020; 2022a), whereas GBIs are not (Tucker et al., 2021; Davies et al., 2023).

**Counterfactual vs. Nullifying Interventions**  Across the board, all counterfactual methods (linear and nonlinear) achieve substantially higher completeness and reliability scores than both nullifying methods (INLP and RLACE). Our empirical evaluation framework is equally applicable to both types of interventions: it uses the same oracle probes and test dataset, and is not biased in favor of one intervention type or the other. Yet, consider INLP and AlterRep: in our implementation, AlterRep uses the *same* classifiers as INLP to perform its intervention, only to carry out a counterfactual rather than nullifying intervention; but AlterRep has peak reliability score $0.8346$ and completeness score of $1.0$, while INLP has $0.4644$ and $0.3503$, respectively. Why should the same set of classifiers using a similar method yield such a different outcome?

One possible explanation is a general critique of causal probing: interventions (both counterfactual and nullifying) operate at the level of latent properties, meaning that they can produce embeddings that do not correspond to any specific input (Geiger et al., 2020; Abraham et al., 2022). For instance, consider the subject-verb agreement task. For input $x = $ "The girl with the keys [MASK] the door", what sentence $x_{Z_c=0}$ would correspond to the nullifying intervention $do(Z_c = 0)$? In English, there

is no corresponding noun where grammatical number is "nullified". For counterfactual interventions $\mathrm{do}(Z_c = z')$, however, there is a natural interpretation of the intervened sentence $x_{Z_c=z'}$: "girl" would be swapped out for "girls".

Given that, unlike the counterfactual case, there are no "nullified inputs" $x_{Z=0}$ that can be used to produce ground-truth "nullified embeddings" $\mathbf{h}^l_{Z=0}$ to train oracle probes, one natural interpretation might be that prediction over nullified embeddings is a form of "oracle probe distribution shift", which would explain the poor performance of nullifying interventions. That is, since oracle probes are never directly trained on intervened embeddings, the fact that they "transfer" better to counterfactual embeddings than to nullified embeddings is the direct result of the lack of correspondence of nullifying interventions to any particular "ground truth" (either in the input or embedding spaces) compared with counterfactual interventions. We argue that such low completeness and reliability scores for nullifying interventions are likely due to this inherent problem with the concept of embedding nullification, rather than any particular intervention methodology.

**Reliability and Task Accuracy**   Figure 2 indicates a clear trend: more reliable and complete interventions show a greater change in BERT's task accuracy (excepting the renegade points whose high change in task accuracy is primarily due to excessive damage to $Z_c$). In particular, the most reliable intervention (AlterRep) consistently shows the greatest change in task accuracy, and the least reliable intervention (i.e., the nullifying interventions INLP and RLACE) show the least clear trend. The GBIs, which are more reliable than INLP and RLACE but less reliable than AlterRep, are capable of damaging task accuracy as much as AlterRep, but only after increasing $\epsilon$ beyond its maximum-reliability setting.

This is an intuitive result: in the case where BERT does indeed perform the task by leveraging its representation of $Z_c$, then more reliable interventions would have a greater effect on the model's task performance. We do not claim that this is necessarily the case (e.g., our results may look different if we intervene in BERT's earlier layers) – rather, we take the clear relationship between between intervention reliability and the change in task accuracy to be a strong indicator that more reliable methods do indeed yield stronger and more consistent results. This finding reinforces the utility of our framework in evaluating the quality of causal probing interventions as tools for studying models' use of latent representations.

## 7   CONCLUSION

In this work, we proposed a general empirical evaluation framework for causal probing, defining the reliability of interventions in terms of completeness and selectivity. Our framework makes it possible to directly compare different kinds of interventions, such as linear vs. nonlinear or nullifying vs. counterfactual methods. We applied our framework to study leading causal probing techniques, finding that they all exhibit a tradeoff between completeness and selectivity. Counterfactual interventions tend to be more complete and reliable, and nullifying techniques are generally more selective. In particular, we find that a linear counterfactual intervention has a very favorable tradeoff between completeness and selectivity, while a closely-related nullifying intervention using the exact same probes has very low reliability. This suggests that underlying differences between interventions types, rather than differences between specific methods of each type, may explain why counterfactual interventions appear more reliable than nullifying methods for causal probing.

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

## A   LIMITATIONS

**Experimental Task and LLM**   In this work, we focus primarily on developing an empirical evaluation framework to evaluate causal probing reliability, and deploy this framework in the context of a relatively simple model (BERT) and task (subject-verb agreement). As we discuss in Section 4, we intentionally select a more straightforward, well-studied experimental setting in order to focus on our framework and the distinctions it reveals between several types of causal probing interventions. That is, despite the proliferation of much larger and more powerful LLMs than BERT, there is still no commonly agreed-upon method for interpreting BERT's learned representations and explaining its behavior, even for simple zero-shot prompting tasks on which it achieves high accuracy. As our focus is to rigorously evaluate existing causal probing methodologies (many of which have been designed specifically with masked language models like BERT in mind), we believe it is more useful to evaluate causal probing methods in the simpler and better-understood context of BERT than it would be to scale our empirical analysis to larger models.

However, it is important to note that our evaluation framework makes no assumptions regarding LLM scale or architecture, nor the complexity of causal task structures. Thus, now that we have demonstrated the effectiveness of our framework in discovering new insights regarding leading intervention methods and providing the first hard evidence to empirically inform long-standing debates regarding causal probing, such as the conceptual issues with nullifying interventions (Abraham et al., 2022; Kumar et al., 2022), we hope that our framework will be deployed and improved upon by future work to study larger and more complex tasks and models, such as autoregressive decoder-only (GPT-style) models and tasks with many more latent properties of interest.

**Multiple Layers**   Another potential limitation of our work is that our experiments only explore interventions in the context of a single layer. As we explain in Section 4, we only examine the final layer in order to prevent information from non-intervened embeddings to be recovered by subsequent attention layers (as observed by Elazar et al. 2021). However, our framework makes no assumptions about the specific layer to analyze – indeed, it is even possible to study the completeness, selectivity, and reliability of interventions performed in earlier layers $l$ with respect to their impact on oracle probes in downstream layers $l' > l$. We recommend such study as an interesting direction for future work, particularly in developing approaches for understanding how information is distributed across different contextualized token embeddings and accessed by downstream attention heads (e.g., as studied in circuit discovery; Wang et al., 2023; Conmy et al., 2023).

**Oracle Probe Approximation**   Finally, in our main paper, we report only results obtained using MLP oracle probes (a decision which we justify at length in Section 4.3). However, we also performed experiments with linear oracle probes, which we report in Appendix D.3. In general, these

results to be similar: the ordering of methods and general trends remain the same, and the main differences are that INLP and RLACE have slightly better completeness and lower selectivity (yielding a marginally higher reliability score) and AlterRep also has lower selectivity (with a similar overall trend). This is not surprising, as linear oracle probes are expected to be more vulnerable to linear interventions than MLPs. Given that our goal in approximating oracle probes is to find the strongest possible probe that is best able to recognize the model's representation, we believe that the results from the MLP oracle probes are more accurate, which is why we focus on them in the main paper.

# B  Framework Details

**Completeness of Nullifying Interventions**  In Equation (2), we define the "goal" distribution $P_Z^*$ of a nullifying causal probing intervention as being the uniform distribution – i.e., for a perfect nullifying intervention, $P_Z^* = P_o(Z \mid \mathbf{h}_{Z=0}^{l^*}) = \mathcal{U}(Z)$. However, this is only true in the case of *causal probing*, not *concept removal*, which is a more common use case for nullifying interventions (see Section 2). That is, in the case of causal probing, the goal of an intervention is to intervene on a model's representation during its forward pass, feeding the intervened embedding back into the model and observing the change in the model's behavior (as described in Section 2). Recall that the purpose of an oracle probe $o$ is to decode model $M$'s representation of a given property $Z$, not to predict its ground truth value – that is, even if $M$ encodes the incorrect value of $Z = z'$ rather than $Z = z$ for a given input, the oracle probe should still decode the incorrect value $Z = z'$. Indeed, this is precisely the principle behind using oracle probes in the case of counterfactual interventions that change the representation of $Z = z$ to counterfactual value $Z = z'$, where oracle probes are used to validate the extent to which the representation has actually been changed to encode this counterfactual value, and the ideal counterfactual intervention yields $P_o(Z = z' \mid \mathbf{h}_{Z=z'}^{l^*}) = P^*(Z = z') = 1$. However, in the case of nullifying interventions $\mathrm{do}(Z = 0)$, an intervened embedding $\mathbf{h}_{Z=0}^{l^*}$ would ideally remove all information encoding $M$'s representation of the value taken by $Z$, meaning that the $M$ would not encode any value $Z = z_1$ as being more probable than $Z = z_2$ (as any information that is predictive of the value taken by $Z$ should have been removed). In this case, the oracle probe $o$ would predict an equal probability $P_o(Z = z_i \mid \mathbf{h}_{Z=0}^{l^*})$ for any given value $z_i$ that may be taken by $Z_i$ – i.e., $P_o(Z \mid \mathbf{h}_{Z=0}^{l^*}) = P_Z^* = \mathcal{U}(Z)$.

However, this is not the case in the context of information removal, where the goal of an intervention $\mathrm{do}(Z = 0)$ is to remove all information that is predictive of $Z$ from embedding representations $\mathbf{h}_{Z=0}^{l^*}$ such that no probe $g$ can be trained to predict $P_g(Z \mid \mathbf{h}_{Z=0}^{l^*})$ any better than predicting $P(Z)$ – i.e., ignoring the embedding entirely and simply mapping every input to the label distribution $P(Z)$ (Ravfogel et al., 2023). In this case, the probe $g$ is trained on intervened embeddings $\hat{\mathbf{h}}_{Z=0}^l$, in which case it can learn to map every such embedding to the label distribution $P(Z)$, which yields superior performance relative to predicting the uniform distribution $\mathcal{U}(Z)$ in any case where the label distribution $P(Z)$ is not perfectly uniform, as such a $g$ would have an expected accuracy equal to the proportion of test instances with the most common label $Z = z_{\mathrm{argmax}}$ (which would be greater than the accuracy $\frac{1}{k}$ expected by defaulting to $\mathcal{U}(Z)$).

The key technical distinction between these two use cases of nullifying interventions is whether or not probes or models are trained or fine-tuned in the context of interventions. In the case of causal probing, they are not – the (frozen) model $M$ has no opportunity to recover the original value of $Z = z$ following a nullifying intervention $\mathrm{do}(Z = 0)$, and this should be reflected by oracle probes. This is natural, given that the purpose of causal probing is to interpret the properties used by $M$ in making a given prediction, not to test whether $M$ can be trained to recover properties removed by interventions; and this is reflected by oracle probes $o$, which are never trained on intervened embeddings. In contrast, for concept removal, probes (or models) are trained on intervened embeddings, and may learn to recover properties removed by interventions, meaning that – even in the worst case where all information has been removed – it would at least be possible to learn to reproduce the label distribution $P(Z)$; but there is no reason to expect a model $M$ or oracle probe $o$ to do so, given that they have never been trained on intervened embeddings. Thus, while we define the "goal" distribution $P_Z^* = \mathcal{U}(Z)$ for measuring the completeness of nullifying interventions as being $\mathcal{U}(Z)$ rather than $P(Z)$, this distribution would instead be $P_Z^* = P(Z)$ in the case of concept removal.

|  | $Z_e = \varnothing$ | $Z_e = $ Sg | $Z_e = $ Pl | Total |
|---|---|---|---|---|
| $Z_c = $ Sg | 176K | 31K | 5K | 213K |
| $Z_c = $ Pl | 78K | 10K | 4K | 92K |
| Total | 254K | 41K | 9K | 305K |

Table 2: **Contingency Table on Test Set**. Distribution of data across combinations of causal and environmental variables. $Z_e = \varnothing$ denotes instances which have no prepositional phrase attached to the subject (and thus, contain no environmental variable). Note that the label distributions are unbalanced: $P(Z_c = $ Sg$) = 69.8\%$ and $P(Z_e = $ Sg $\mid E \neq \varnothing) = 81.5\%$.

## C EXPERIMENTAL DETAILS

### C.1 DATASET

### C.2 MLP PROBES

We do a hyperparameter sweep with grid search for the MLP probes (both oracle and interventional) that we train. The hyperparameters we consider are:

- Num. layers: [1, 2, 3]
- Layer size: [64, 256, 512, 1024]
- Learning rate: [0.0001, 0.001, 0.01]

Since the MLPs are performing classification, they are trained with standard cross-entropy loss. The probes are trained for 8 epochs, and the best probe is selected based on validation accuracy.

### C.3 INTERVENTIONS

**Gradient Based Interventions:** For all gradient-based intervention methods (Davies et al., 2023), we define the maximum perturbation magnitude of each intervention as $\epsilon$ (i.e., $||\hat{\mathbf{h}}_Z^l - \mathbf{h}^l||_\infty \leq \epsilon$), and experiment over a range of $\epsilon$ values between 0.005 to 5.0 – specifically, $\epsilon \in [0.005, 0.006, 0.007, 0.009, 0.011, 0.013, 0.016, 0.019, 0.024, 0.029, 0.035, 0.042, 0.051, 0.062, 0.076, 0.092, 0.112, 0.136, 0.165, 0.2, 0.286, 0.409, 0.585, 0.836, 1.196, 1.71, 2.445, 3.497, 5.0]$. (These are the points along the x-axis for the results visualized in Figures 1 and 4. Figures 3 and 9.) We consider the following gradient attack methods for GBIs:

1. **FGSM** We implement Fast Gradient Sign Method (FGSM; Goodfellow et al., 2015) interventions as:
$$h' = h + \epsilon \cdot \text{sgn}\left(\nabla_h \mathcal{L}\left(f_{\text{cls}}, x, y\right)\right)$$

2. **PGD** We implement Projected Gradient Descent (PGD; Madry et al., 2017) interventions as $h' = h^T$ where
$$h_{t+1} = \Pi_{\mathcal{N}(h)}\left(h_t + \alpha \cdot \text{sgn}\left(\nabla_h \mathcal{L}(f_{\text{cls}}, x, y)\right)\right)$$
for iterations $t = 0, 1, \ldots, T$, projection operator $\Pi$, and $L_\infty$-neighborhood $\mathcal{N}(h) = \{h' : \|h - h'\| \leq \epsilon\}$. For PGD, we use 2 additional hyperparameters: iterations $T$ and step size $\alpha$, while fixing $T = 40$, as suggested by (Davies et al., 2023).

3. **AutoAttack** AutoAttack (Croce & Hein, 2020) is an ensemble of adversarial attacks that includes FAB, Square, and APGD attacks. Auto-PGD (APGD) is a variant of PGD that automatically adjusts the step size to ensure effective convergence. The parameters used were set as norm $= L_\infty$ and for Square attack, the n_queries=5000.

**Nullifying Interventions:** For nullifying interventions, we project embeddings into the nullspaces of classifiers. Here, the the rank $r$ corresponds to the dimensionality of the subspace identified and erased by the intervention, meaning that the number of dimensions removed is equal to the rank.[6]

---

[6]This is only true for binary properties $Z$ – for variables that can take $n$ values with $n > 2$, the number of dimensions removed is $n \cdot r$.

We experiment over the range of values $r \in [0, 1, ..., 40]$. (These are the points along the x-axis for the results visualized in Figures 5 and 11.) We consider the following nullifying interventions:

1. **INLP** We implement Iterative Nullspace Projection (INLP; Ravfogel et al., 2020) as follows: we train a series of classifiers $w_1, ..., w_n$, where in each iteration, embeddings are projected into the nullspace of the preceding classifiers $P_N(w_0) \cap \cdots \cap P_N(w_n)$. We then apply the combined projection matrix to calculate the final projection where $P := P_N(w_1) \cap \cdots \cap N(w_i)$, $X$ is the full set of embeddings, and $X_{\text{projected}} \leftarrow P(X)$.

2. **RLACE** We implement Relaxed Linear Adversarial Concept Erasure (R-LACE; (Ravfogel et al., 2022a)) which defines a linear minimax game to adversarially identify and remove a linear bias subspace. In this approach, $\mathcal{P}_k$ is defined as the set of all $D \times D$ orthogonal projection matrices that neutralize a rank $r$ subspace:

$$P \in \mathcal{P}_k \leftrightarrow P = I_D - W^\top W$$

The minimax equation is then solved to obtain the projection matrix $P$ which is used to calculate the final intervened embedding $X_{projected}$, similar to INLP

$$min_{\theta \in \Theta} max_{P \in \mathcal{P}_k} \sum_{n=1}^{N} \ell\left(y_n, g^{-1}\left(\theta^\top P x_n\right)\right)$$

The parameters used for $P$ and $\theta$ included a learning rate of 0.005 and weight decay of 1e-5.

**AlterRep** We implement AlterRep (Ravfogel et al., 2021) by first running INLP, saving all classifiers, and using these to compute rowspace projections that push all embeddings to the positive $Z = \texttt{Pl}$ or negative $Z = \texttt{Sg}$ side of the separating hyperplane for all classifiers. That is, we compute

$$\hat{\mathbf{h}}^l_{Z=\texttt{Sg}} = P_N(\mathbf{h}^l) + \alpha \sum_{w \in \mathbf{W}} (-1)^{SIGN(w \cdot \mathbf{h}^l)} (w \cdot \mathbf{h}^l) \mathbf{h}^l$$

$$\hat{\mathbf{h}}^l_{Z=\texttt{Pl}} = P_N(\mathbf{h}^l) + \alpha \sum_{w \in \mathbf{W}} (-1)^{1-SIGN(w \cdot \mathbf{h}^l)} (w \cdot \mathbf{h}^l) \mathbf{h}^l$$

where $P_N$ is the nullspace projection from INLP.

# D ADDITIONAL RESULTS

## D.1 RELIABILITY BY INTERVENTION HYPERPARAMETER

In Figure 3, Figure 4, and Figure 5, we observe that increasing the degree of control that interventions have over the representation of the target property by increasing the intervention hyperparameter associated with a given intervention type (i.e., $\epsilon, \alpha,$ or rank) generally leads to both improved completeness and decreased selectivity.

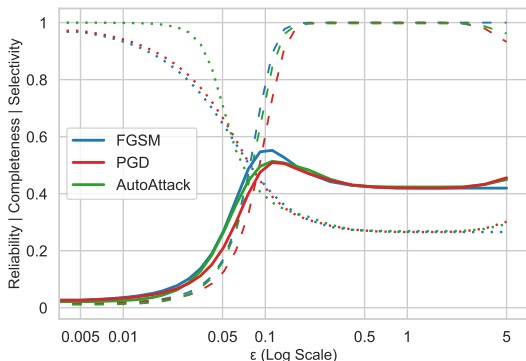

Figure 3: **Counterfactual GBIs.** Reliability (solid), completeness (dashed), & selectivity (dotted) for FGSM, PGD, and AutoAttack targeted at $Z_c$.

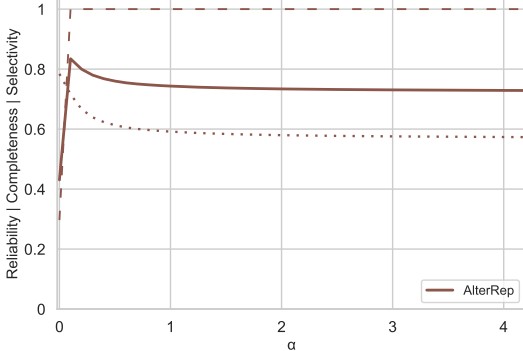

Figure 4: **AlterRep.** Reliability (solid), completeness (dashed), & selectivity (dotted) for AlterRep targeted at $Z_c$.

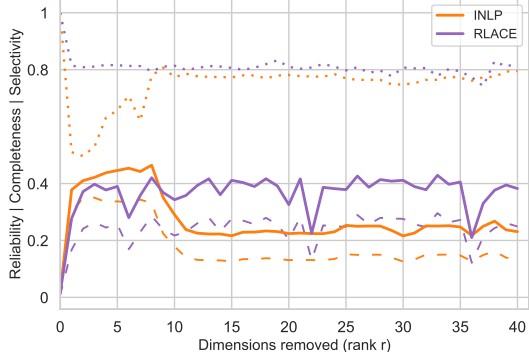

Figure 5: **Nullifying linear methods.** Reliability (solid), completeness (dashed), & selectivity (dotted) for INLP and RLACE targeted at $Z_c$.

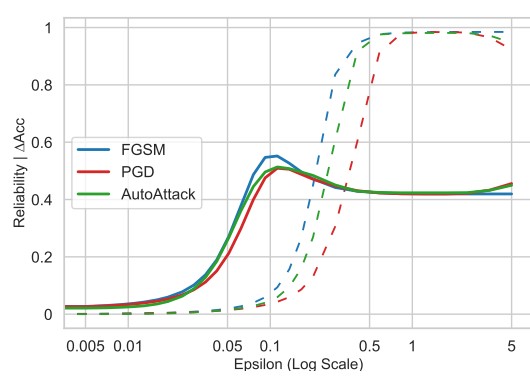

Figure 6: **Task accuracy for counterfactual GBIs.** Reliability (solid) and task accuracy (dashed) for FGSM, PGD, and AutoAttack targeted at $Z_c$.

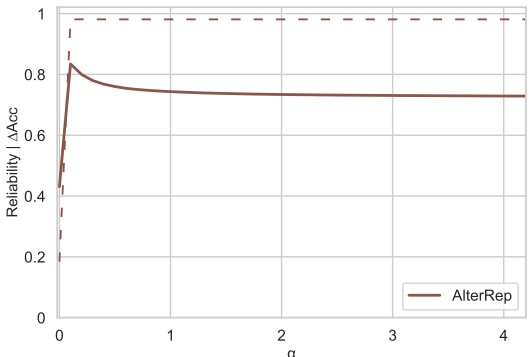

Figure 7: **Task accuracy for AlterRep.** Reliability (solid) and task accuracy (dashed) for AlterRep targeted at $Z_c$.

### D.2 TASK ACCURACY BY INTERVENTION HYPERPARAMETER

Figure 6, Figure 7, and Figure 8 show reliability and task accuracy at various hyperparameter values for each method. Generally, increasing the hyperparameter values (and hence "amount of damage") results in more severe change in BERT's task accuracy.

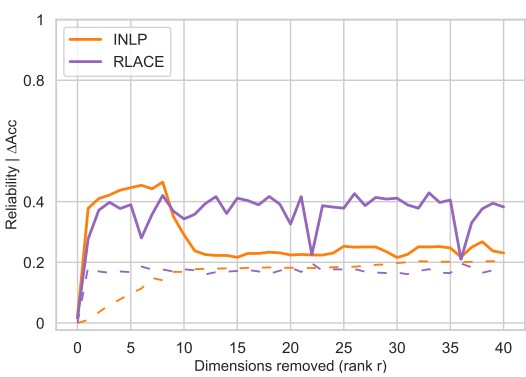

Figure 8: **Task accuracy for nullifying linear methods.** Reliability (solid) and task accuracy (dashed) for INLP and RLACE targeted at $Z_c$.

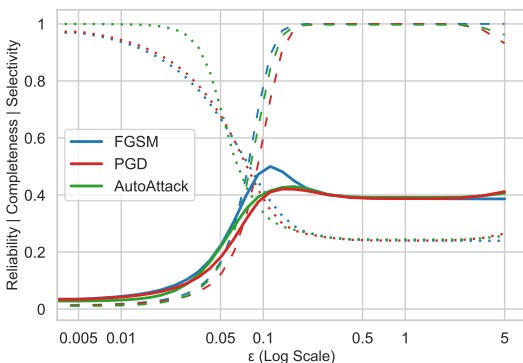

Figure 9: **Linear oracle probe on counterfactual GBIs.** Reliability (solid), completeness (dashed), & selectivity (dotted) for FGSM, PGD, and AutoAttack targeted at $Z_c$, using *linear* oracle probes for evaluation.

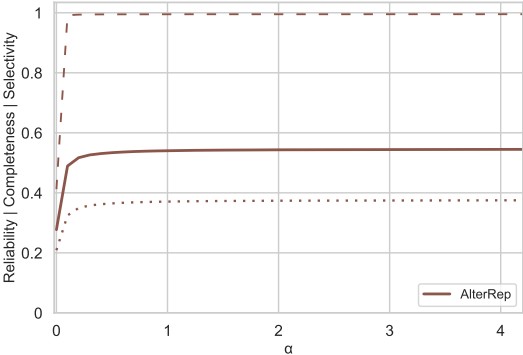

Figure 10: **Linear oracle probe on AlterRep.** Reliability (solid), completeness (dashed), & selectivity (dotted) for AlterRep targeted at $Z_c$, using *linear* oracle probes for evaluation.

### D.3 LINEAR ORACLE PROBES

In this section, we present plots depicting reliability, completeness, and selectivity where the oracle probe has a *linear* architecure (not MLP) for each intervention. These linear probes are trained with cross-entropy loss, and a grid search was performed over learning rates in [0.0001, 0.001, 0.01, 0.1] to find the hyperparameter with the lowest validation-set accuracy. The plots are shown in Figures 9, 11, and 10.

For counterfactual GBIs, the linear oracles present very similar results to those computed with MLP oracles, shown in Figure 3. AlterRep also shows similar results to those presented in 4, except that selectivity is lower: this is expected, as linear oracle probes should be expected to be less resilient to linear interventions than MLP oracle probes. INLP and RLACE show a lower selectivity and higher completeness when evaluated with linear oracle probes versus MLP oracles, for the same reason.

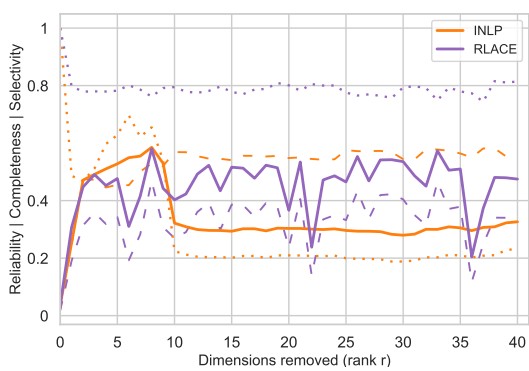

Figure 11: **Linear oracle probe on nullifying linear methods.** Reliability (solid), completeness (dashed), & selectivity (dotted) for INLP and RLACE targeted at $Z_c$, using *linear* oracle probes for evaluation.

