# OpenReview forum: "Measuring the Reliability of Causal Probing Methods: Tradeoffs, Limitations, and the Plight of Nullifying Interventions"
_ICLR.cc/2025/Conference — Submitted to ICLR 2025_

### Official Review · Reviewer_HqKg · 2024-11-01

**Soundness:** 2
**Presentation:** 2
**Contribution:** 3
**Rating:** 6
**Confidence:** 3

**Summary:**

The authors propose metrics to measure the completeness and selectivity of causal probing techniques, and conduct an empirical study with diverse causal probing techniques; including recently proposed counterfactual intervention techniques. The motivation behind the empirical study is to formally understand the challenges with causal probing, and establish trends to understand the relative advantages/disadvantages of the different causal probing techniques; counterfactual vs nullifying intervention techniques, and linear vs non-linear counterfactual intervention techniques. The authors report results with BERT on the LGD subject-verb agreement dataset, and find that counterfactual probing methods are better than nullifying probing methods in terms of completeness, and also linear counterfactual techniques fare better than non-linear counterfactual techniques in terms of reliability.

**Strengths:**

- The proposed evaluation framework is novel to the best of my knowledge, and the analysis of recently proposed counterfactual intervention techniques has not been done in prior works. However, I am not very familiar with the literature in causal probing, hence I am not the best judge for this.

- The experiment results are present clearly and there is a good discussion around them, which makes it easy to follow the main claims. The explanations given in the paper behind observed trends are good and justified.

- The observation of counterfactual interventions better than nullifying interventions for causal probing maybe of significance to the community, and help in development of better causal probing techniques.

- The experiment setup is comprehensive enough as it covers a variety of causal probing techniques, and considers widely used benchmark for causal probing.

**Weaknesses:**

- There should be more details provided regarding the performance of (trained) oracle probe. While the authors just report that the oracle obtains high accuracy for predicting the causal variables, more details are needed regarding the distribution learnt by the oracle probe. For example, the completeness metric for nullifying interventions (e.q. 2)  is designed under the assumption that the oracle probe has a uniform  distribution after the null intervention. Is this true for the oracle probe trained by the authors in their empirical study? If this is not the case and the distribution learnt by oracle probe is not uniform, that might bias the completeness metric for nullifying interventions, leading to the question whether we can trust the observed trend that nullifying interventions are worse than counterfactual interventions for completeness. Hence, the authors should verify the oracle distribution in the case of nullifying interventions and check how similar it is to the uniform distribution.

- Table 1 should report standard error/deviation as well over the different examples in the test set. Same comment for figures, but if the error bars are not too significant then they can be dropped, but they must be explicitly mentioned in the appendix. Also, it is not clear how many random seed were used for conducting the empirical study. Are the findings based on a single random seed? If that is the case then I advise the authors to conduct experiments with multiple random seeds to have robust trends.

- Overall, I have some concerns with the writing of the paper in terms of details regarding notations and metrics. For example, for the metrics (e.q. 1, 2, 3), why do the authors remove the dependence on $z_{i'}$ from $P^\star$ and $\hat P$, as they are a function of the value set after intervention. Similarly, the derivation behind the metric are not clearly stated in some cases. For example, equation 3, what is the min and max of the distributions $P^\star$ and $\hat P$? Over what argument are we taking the min/max over? More justification should be provided for the derivation of the selectivity metric. I have expanded more on this point about notations etc. in the questions section ahead.

**Questions:**

- Regarding the discussion on linear vs non-linear counterfactual probing methods, can the authors conduct a small experiment where the intervention happened on not the last layer? This will help to understand whether the improvement in completeness with linear counterfactual probing is due to intervention happening on the last layer, or there are some other reasons behind it. Also, can you provide an example where non-linear counterfactual probing is better than linear counterfactual probing?

- The training objective of the oracle probe is not clear to me, please provide more details on it.

- The notation of $\hat h^{l}$ and $h^{l^{\star}}$ should be more clearly defined in the text. Also, why are authors using $h^{l^{\star}}$ and not $h^{\star^l}$, was there some specific reason? Currently its a bit confusing as the notation for layers is not consistent, $l$ versus $l^{\star}$ in $\hat P$ and $P^{\star}$.


Minor comments

- It might be good to have a figure denoting the overall design of the evaluation framework.

- The contributions in the introduction can be stated more clearly; either in terms of bullet points and some of the content about the limitation of causal probing can be brought in the introduction.

- There is a typo on linear 172; it should be $c(\hat h^{l, k})$ instead of $c(\hat h^{l, i} )$.

- Line 64, it would be good to provide a clear reference to prior works that criticize the nullifying interventions; instead of referring to Section 2.

---

> ### Author Response · Authors · 2024-11-21
> **Rebuttal (part 1/N)**
>
> Thank you for your detailed read of our paper and providing this comprehensive and thoughtful review. We appreciate that you found our framework to be novel, our results to be clear and justified, our findings to be potentially significant to the community, and our experiments to be comprehensive.
>
> We address each of the concerns you raised as follows:
>
> **C1: Target distribution of oracle probes**
>
> You suggested that if “the distribution learnt by oracle probe is not uniform, that might bias the completeness metric for nullifying interventions [...] Hence, the authors should verify the oracle distribution in the case of nullifying interventions and check how similar it is to the uniform distribution.”
>
> In our paper, we use oracle probes to evaluate interventions by first observing the distribution emitted by oracle probes for intervened embeddings, then assessing how similar this distribution is to the target distribution (as defined in Section 3). However, if – as you suggested – this evaluation process was further “verified” by measuring how similar the oracle probe distribution is to the target (uniform) distribution, this would amount to “verifying” the evaluation process *based on how high a score it assigns to the interventions it is evaluating*, which would be circular reasoning.
>
> Please let us know if we have misunderstood your suggestion, and we would be happy to discuss this matter further.
>
> **C2: Standard deviation and random seed**
>
> Below, we report the [standard error](https://en.wikipedia.org/wiki/Standard_error#Exact_value) for each result in Table 1:
>
> |            | completeness    | selectivity     | reliability     | optimal hyperparameter |
> |------------|-----------------|-----------------|-----------------|------------------------|
> | FGSM       | 0.8923 ± 0.0011 | 0.3994 ± 0.0018 | 0.5518 ± 0.0017 | $ \epsilon $  = 0.112  |
> | PGD        | 0.7343 ± 0.0016 | 0.3897 ± 0.0018 | 0.5092 ± 0.0016 | $ \epsilon $  = 0.112  |
> | AutoAttack | 0.8433 ± 0.0013 | 0.3692 ± 0.0019 | 0.5136 ± 0.0018 | $ \epsilon $  = 0.112  |
> | AlterRep   | 1.0000 ± 0.0000 | 0.7842 ± 0.0017 | 0.8346 ± 0.0012 | $ \alpha $  = 0.1      |
> | INLP       | 0.3308 ± 0.0013 | 0.7792 ± 0.0012 | 0.4644 ± 0.0013 | r = 8                  |
> | RLACE      | 0.2961 ± 0.0013 | 0.7782 ± 0.0012 | 0.4290 ± 0.0014 | r = 33                 |
>
> As for the figures in the main paper, which are all scatterplots, it is not clear how we would report standard deviation in such plots. However, these figures are based on results that can also be visualized as line plots (across a greater number of figures) that are included in the appendix, where Figure 1 corresponds to Figures 3-5.
>
> Thus, we have recomputed these figures with standard error visualized as a semi-transparent region around each line in the plot, and make them available at the following anonymized links for [GBIs](https://osf.io/ag9sx?view_only=b9189a028ce5434cbf4ece37ca8e651b), [AlterRep](https://osf.io/jdmhs?view_only=b9189a028ce5434cbf4ece37ca8e651b), and [nullifying methods](https://osf.io/rf9u3?view_only=b9189a028ce5434cbf4ece37ca8e651b) (corresponding to Figures 3, 4, and 5, respectively).
>
> As observed in the included table and all figures, the standard error is very small, and is barely visible in visualizations. This is due to the very large size of the test set, which includes 50K instances in total. We will add these error values to Table 1 and Figures 3-5.
>
> Regarding the random seed: we split the full train/val/test set from the LGD subject-verb agreement task data only once using a single random seed, as is standard; but we do train several different randomly-initialized interventional probes and approximated oracle probes across various hyperparameter settings ($n = 36$) for each experiment, each time selecting the probes with the highest validation-set accuracy. (See Appendix C.2 for further details.)
>
> (Continued in rebuttal part 2/N.)

---

> ### Author Response · Authors · 2024-11-21
> **Rebuttal (part 2/N)**
>
> (Continued from rebuttal part 1/N.)
>
> **C3: Notation**
>
> Thank you for noticing two typos in our notation: $Z$ was incorrectly removed as a subscript for $\hat{P}_Z$ in equations 1, 2, 3, and some of the surrounding text; and $i$ should be replaced with $k$ in the superscript of line 172. We will correct both typos in our next draft.
>
> As for computing min/max over distributions: we take the maximum probability in each corresponding distribution (i.e., max, not argmax). E.g., for $P(Z = z_1) = 0.3, P(Z = z_2) = 0.7$, these operations would yield $min(P) =0.3$ and $max(P) = 0.7$. The only reason this is required in Equation 3 is to normalize the selectivity metric such that it is bounded by [0, 1] (see line 200).
>
> Regarding $\hat{h}^l$ and $h^{l^*}$: as described in Section 3, $\hat{h}^l$ is the embedding obtained following the application of an intervention method that is being evaluated, and $h^{l^*}$ is a hypothetical embedding that would be obtained following the application of an idealized “perfect” intervention (which we do not have access to in practice). We agree with your suggestion that $h^{l^*}$ would be better denoted as $h^{*^l}$, and we will modify our notation correspondingly.
>
> **C4: Unclear training objective of approximated oracle probe**
>
> We explain how approximated oracle probes are trained in Section 4.3 (lines 282-290) and Appendix C.2: specifically, the training objective is to minimize the cross-entropy loss over the value taken by the property given an embedding input. Please let us know if you have any additional questions about the content in these sections, or if there are specific details we have not included that you would like us to add.
>
> **C5: Interventions on earlier layers**
>
> First, we will clarify why we only experimented on final-layer embeddings in our paper: as described in Section 4.2 (lines 265-269), we only consider interventions over final-layer embeddings in our experiments to ensure that our results are not impacted by the recoverability phenomenon observed by Elazar et al. (2021), allowing for direct analysis of the impact on LM task accuracy (Sections 4.5 and 5.2).
>
> However, our framework can easily be applied to any layer. Thus, as requested, we have performed an additional followup experiment over a few earlier layers $l \in \{ 1, 4, 7, 10\}$ and the two top-performing counterfactual interventions (AlterRep and FGSM GBIs) to compare with the existing results for the final layer $l = 12$ and study the difference between linear (AlterRep) and nonlinear (FGSM GBIs) methods.
>
> The results are visible at [this anonymized link](https://osf.io/x8tym?view_only=b9189a028ce5434cbf4ece37ca8e651b), where we plot the reliability (solid line), completeness (dashed line), and selectivity (dotted line) of FGSM and AlterRep by layer, using the intervention hyperparameter values for each layer that yield the maximum reliability (cf. Table 1). These results are obtained by following the same procedure as all experiments in the main paper, where the only element that has changed is the layer under investigation.
>
> These results show that, as hypothesized (see lines 460-462), the nonlinear counterfactual method (FGSM) performs better in earlier layers, and the linear counterfactual method (AlterRep) performs better in later layers. This indicates that the representation of grammatical number ($Z_c$) is nonlinearly encoded in earlier layers, but becomes increasingly linear in later layers (cf. Alain & Bengio, 2017).
>
> **C6: Provide an example where non-linear causal probing is better than linear causal probing**
>
> Theoretically, in any case where properties are encoded nonlinearly – as may be expected of highly nonlinear objects such as deep neural networks – nonlinear probes will be better able to recognize these representations, as is often observed empirically (Pimentel et al., 2020; Belinkov et al., 2022; Davies et al., 2023). In such cases, causal probing interventions operating over nonlinear probes would be able to perform more *complete* interventions on a neural network’s representation of the target property, as we observe in the results reported for **C5** above. However, some linear causal probing methods like AlterRep or INLP come with, e.g., stronger theoretical guarantees on “collateral damage”, which can lead to a more favorable tradeoff in favor of selectivity even if interventions are not fully complete (see Table 1).
>
> **C7: Include a figure denoting overall framework and contributions bullets in the intro**
>
> We agree that these suggestions would improve the readability of our paper and the clarity of our contribution. We will make both of the recommended additions. Likewise, we will implement the suggestion to add some citations on line 64 that are currently not cited until Section 2.
>
> **Conclusion**
>
> We hope we have resolved all of your concerns. If not, we remain available for further discussion.

---

> ### Comment · Reviewer_HqKg · 2024-11-25
>
> Thanks a lot for your clear and detailed response! A lot of concerns were addressed, but I am still not sure about the distribution of oracle probe. I don't understand the "circular argument" pointed by the authors regarding my concern about the oracle probe for nullifying interventions. From my understanding, the oracle probe gives a distribution over whether the property $Z=z$ holds after an intervention. So authors have access to $p_Z(z|h_{Z}^l)$, and they can test whether it is a uniform distribution or not. As the completeness metric for nullifying interventions (eq 2) assumes it to be uniform and then checks how much does $\hat p_Z(z|h_{Z}^l)$ differs from it. But if $p_Z(z|h_{Z}^l)$ is not uniform to being with, then this metric might be biased. Please let me know if you need more clarification on this.

---

> > ### Author Response · Authors · 2024-11-27
> > **Response to "Official Comment by Reviewer HqKg"**
> >
> > Thank you for your response to our rebuttal! We are glad that we have addressed most of your concerns, and will do our best to address the remaining question you raised.
> >
> > The source of confusion may be the following: you stated that “authors have access to $p_Z(z | h)$”. We do not, as this would require a perfect oracle probe $o$, which does not exist in practice. What we do have access to is an approximation of $p_Z(z | h)$ emitted by approximated oracle probes $\hat o$. Similarly, we do not have access to “ground-truth” nullified embeddings $h^*_{Z=0}$, as they do not exist – i.e., there exists no English-language text input $x_{Z=0}$ that would yield a nullified embedding $h^*_{Z=0}$, as discussed in lines 484-489. The only nullified embeddings we have are those obtained using the methods under investigation, $\hat h_Z$; and the only oracle probes we have are approximations $\hat o$. Thus, without access to ground-truth $h^*_{Z=0}$, it is impossible to directly test whether approximated oracle probes $\hat o$ yield a uniform distribution $p_Z(z | h^*_{Z=0}) = \mathcal{U}(Z)$.
> >
> > More generally, given (a) nullified embeddings $\hat h_Z$ obtained using existing nullifying methods, and (b) approximated oracle probes $\hat o$, there are only two non-circular approaches to validate one using the other:
> > - Option 1 ($a \to b$): if we wish to validate approximated oracle probes using nullified embeddings, we must assume (or provide some independent evidence) that nullifying interventions produce (nearly) perfect approximations of nullified embeddings.
> > - Option 2 ($b \to a$): if we wish to validate nullifying interventions using approximated oracle probes, then we must assume (or provide some independent evidence) that the approximated oracle probes are (nearly) perfect approximations.
> >
> > That is, without first assuming that either (a) or (b) is a sufficiently high-quality approximation, the validation procedure you proposed would indeed be circular. Our experiments follow Option 2 ($b \to a$), assuming near-perfect oracle probes instead of near-perfect nullifying interventions; but we do not do so without evidence. That is, while there are no external, non-circular approaches to validating interventions (as our framework is the first to do so), there *are* such approaches to validating approximated oracle probes, such as measuring accuracy on a held-out ground-truth test set (which is 99.4% for $Z_c$ and 88.4% for $Z_e$ — see line 377). This validation procedure is an imperfect proxy for addressing your question of whether the uniform distribution is the best target for nullifying interventions, as it only tests the quality of the oracle probe approximation $\hat o$ for ground-truth embedding/label pairs $(h, z)$, and not on whether it emits a uniform distribution for the hypothetical perfect nullified embeddings $h^*_{Z=0}$; but as explained above, this is impossible to test in practice.
> >
> > Please let us know if we have fully addressed your concern, or if you have any remaining questions. We remain available for further discussion.

---

> > > ### Comment · Reviewer_HqKg · 2024-12-02
> > >
> > > Thanks for engaging in discussion! My concerns have been addressed and I have increased my score.

---

> > > > ### Author Response · Authors · 2024-12-04
> > > >
> > > > Dear Reviewer,
> > > >
> > > > Thank you very much for engaging in productive dialogue with us and updating your assessment accordingly! We sincerely appreciate your detailed and thoughtful review, which has been very helpful for us in suggesting important revisions and key followup experiments (whose results we discussed in our rebuttal).
> > > >
> > > > Thank you,
> > > >
> > > > Authors

---

### Official Review · Reviewer_ynad · 2024-11-03

**Soundness:** 3
**Presentation:** 4
**Contribution:** 3
**Rating:** 5
**Confidence:** 3

**Summary:**

How do you evaluate a causal probe when you can't even be sure it has representation for the right concept (instead of some correlated concept)? This paper seeks to answer this popular question by training "oracle probes" for both the target and off-target concepts, thereby increasing confidence that an intervention to the representation space is *actually* the one that was intended.

With these oracle probes in hand, the authors then propose evaluating causal probing methods according to two desiderata which they formally define: completeness and selectivity (corresponding to whether the representation captures *all* of the concept, and whether it *does affect anything else*, intuitively).

These are then used to experimentally evaluate different types of causal probing methods, thereby retrodicting the observation that nullifying interventions are less complete than counterfactual interventions.

**Strengths:**

The proposed method of training "oracle probes" is simple yet versatile: it seems applicable to any attribute with a supervised dataset.
To my knowledge, such an approach is novel, as probes are often used as a source of explanation, rather than as a means to assess the effect of an intervention. The contribution of "oracle probes" thus seems analogous to the development of synthetic controls for causal inference: use a machine learning model to close the causal inference loop, under some suitable assumptions (which here are missing, see weaknesses).

This paper emphasizes that completeness and selectivity empirically induce a trade-off, and so aptly propose the harmonic mean as a way to aggregate these appropriately.

**Weaknesses:**

1. This approach requires enumerating all off-target concepts in order to construct the oracle probe for Z_e. However, this is a common assumption in the causal-inference-in-text literature, so not surprising.
2. A crux of this paper is the development of *good* oracle probes. However, there is very little analysis of how to evaluate the oracle probes (which are then used to evaluate causal probing methods).
3. Currently, this paper only considers binary concepts. This is a common assumption, so not very limiting.

**Questions:**

1. Could you elaborate on the implicit assumptions on the oracle probes? For instance, in the experiments section, you mention the assumption that they are trained in a such a way that they have no spurious correlation in their training data. Are there other theoretical assumptions on the oracle probes?

UPDATE Dec 8, 2024: I would like to thank the authors for their thorough responses to my review as well as the other reviewers, and apologies for missing the discussion period for this paper. At the AC's request I am now updating my review, having read all other reviews and rebuttals.

The authors brought up a good point in C1, that new oracles can be added independently of other probes. While this doesn't alleviate the need to enumerate the off-target concepts, it does suggest that the test for specificity might be iteratively approximated by adding more and more oracles.

However, fundamentally this approach relies on the validity of the oracle probes, which should be either theoretical or empirical (ideally both).
- Reviewer hJpz and I both highlighted that the paper lacks any theoretical validation for the oracles; the authors acknowledged in the rebuttals that indeed no theoretical assumptions are listed.
- As for empirical evidence of the validity of the oracle probes, the authors in their rebuttal to my review say "we simply perform standard hyperparameter grid search over possible oracle probes and select the MLP architecture and learning rate that yields the highest validation-set accuracy (see Section 4.3 and Appendix C.2)". The key issue here is that validation-set accuracy cannot falsify whether the oracle probes have errornously picked up on a spuriously-correlated concept, because the validation set is sampled from the same distribution as the training. An empirical validation for the oracles probes should demonstrate that oracle O_i generalizes out of distribution, changing Z_i and no other concepts Z_j. Validation set accuracy does not test any of these.

Due the the heavy reliance on the validity of the oracle probes, the lack of either theoretical or empirical validation, I am slightly lowering my score. I believe this approach is promising but does not yet provide clear conclusions without further oracle validation.

---

> ### Author Response · Authors · 2024-11-21
> **Rebuttal**
>
> Thank you for your thoughtful and comprehensive review. We appreciate that you found our approach to be simple, versatile, and novel.
>
> We address each of the concerns you raised as follows:
>
> **C1: Must enumerate all probed properties**
>
> While it is indeed necessary to enumerate the set of properties $Z_1, …, Z_k$ to be used in computing selectivity, each corresponding approximated oracle probe $o_1, …, o_k$ can both be trained and applied to evaluate interventions at test-time fully independently of the others. This means that it is possible to add new properties post-hoc, train oracles on only the new properties, and easily recompute the aggregated selectivity without having to re-train existing oracle probes.
>
> **C2: How should oracle probes be evaluated?**
>
> It is true that our framework crucially relies on high-quality oracle probe approximations, and the question of how best to evaluate such approximations is indeed an open one. In our experiments, we simply perform standard hyperparameter grid search over possible oracle probes and select the MLP architecture and learning rate that yields the highest validation-set accuracy (see Section 4.3 and Appendix C.2). We opted for this approach in accordance with Occam’s razor: as we are the first to define and experiment with oracle probes, we selected from the range of possible oracle probes in the way that one would select any other model or probe (i.e., the one with the best performance on the validation set).
>
> However, we also experimented with linear oracle probes (see Appendix D.3). These linear oracles yielded similar results to those obtained using the MLP oracles whose results are listed in the main paper – e.g., all major findings are the same – which gives us some confidence that different types of oracle probe approximations generally yield similar results in the experimental setting explored in this work. However, we acknowledge that this is a purely empirical observation made on the basis of a limited set of experiments, and it is possible that the matter of oracle probe selection could be much more consequential in more complex experimental scenarios. This is an important consideration for future work, and we will add further discussion on this point to the paper.
>
> **C3: Experiments consider only binary properties**
>
> It is true that our experiments consider only binary properties, which – as you noted – is a standard setting explored in the literature, and thus is a natural place to begin validating the proposed causal probing evaluation framework. However, our framework (as provided in Section 3) is explicitly defined to accommodate properties with more than two possible values (see lines 178-180).
>
> **C4: Are there additional implicit assumptions for oracle probes?**
>
> No, we enumerate the full set of *experimental assumptions* for oracle probe approximations in Section 4.3 (lines 282-290) – i.e., that they should be trained on a disjoint set from interventional probes, that the target property should be conditionally independent of the other properties, and that the label distribution of the target property should be preserved while enforcing conditional independence. However, please note that these are not *theoretical assumptions* for defining oracle probes – rather, they are empirical measures to obtain more reliable oracle probe approximations (see lines 285-290).
>
> We will better clarify this point in Section 4.3 – specifically, that these experimental assumptions are introduced as initial “best practices”, and that future work could experiment with additional measures to improve such approximations.
>
> **Conclusion**
>
> We hope we have resolved all of your concerns. If not, we remain available for further discussion.

---

> > ### Author Response · Authors · 2024-11-27
> > **Request response to our rebuttal**
> >
> > Once again, we thank you for your initial review of our paper. To promote a productive discussion period, we would greatly appreciate it if you are able to respond to our rebuttal.
> >
> > Have we fully resolved all of your concerns, or do you have any additional questions? We are more than happy to clarify any potential points of confusion and engage in further discussion.

---

> > > ### Author Response · Authors · 2024-12-04
> > > **Please take rebuttal into consideration during meta-review**
> > >
> > > Dear Reviewer,
> > >
> > > We appreciate your initial review of our paper, and regret that we did not receive a response to our rebuttal (posted Nov 20). We hope that our rebuttal fully addressed your concerns.
> > >
> > > Although we were unable to engage with you during the discussion period, we would like to kindly request that you please update your assessment during the meta-review process based on the content of our rebuttal. If we were able to resolve your initial concerns, we hope that you will take this into account.
> > >
> > > Thank you,
> > >
> > > Authors

---

### Official Review · Reviewer_vDrB · 2024-11-04

**Soundness:** 3
**Presentation:** 2
**Contribution:** 2
**Rating:** 5
**Confidence:** 2

**Summary:**

The paper introduces an empirical framework to evaluate causal probing methodologies in language models, particularly focusing on completeness and selectivity. The study highlights a trade-off between these two criteria, showing that counterfactual interventions are generally more effective in altering targeted representations without impacting unrelated properties, unlike nullifying interventions.

**Strengths:**

1. **Novel Evaluation Framework**: I appreciated the proposed structured framework for measuring the reliability of causal probing interventions. Such a standardized idea can be used to evaluate and compare various methods.

2. **Detailed Experimental Analysis**: The experiments are conducted with detailed analysis across multiple intervention methods and settings, offering a comprehensive evaluation of the methods' performance under varied conditions.

**Weaknesses:**

1. **Limited Model Scope**: The major concern is insufficient experimental verification. As the authors also claim in the paper, this evaluation is conducted only on a simple model and task (line 672), which might limit the generalizability of the findings to more complex language models and tasks. Since some ideas seem challenging to achieve in practice (line 145), it would be beneficial to see verification or discussion on how the proposed framework could adapt to more general and complex scenarios.

2. **Redundant Descriptions**: Some sections, like Sections 4 and 5, provide extensive details that could be condensed, with some specifics better suited for an appendix. Simplifying these sections (maybe at the same time introducing more settings and results) could enhance readability without compromising clarity.

**Questions:**

Can the authors provide any guidance or verification on applying the proposed idea to a more general setting? Although the framework appears novel, its current verification relies on several strict assumptions (e.g., full determinism, binary variables, etc.). I’m curious about how this framework could be implemented in a more universally applicable manner.

---

> ### Author Response · Authors · 2024-11-21
> **Rebuttal (part 1/N)**
>
> Thank you for your thoughtful review. We appreciate that you found our framework to be novel and consider our analysis to be detailed and comprehensive across multiple methods and experimental settings.
>
> We address each of the concerns you raised as follows:
>
> **C1: Limited scope of models and tasks: how can the framework be adapted to more general/complex scenarios?**
>
> It is important to distinguish between our framework and our experiments. Our framework, as presented in Section 3, is general with respect to arbitrary language models and tasks – i.e.:
> - Our framework does not make any assumptions regarding language model architecture, parameter count, training distribution, etc.; nor does it make any specific assumptions about task structure. For example, the “causal vs environmental properties” we discuss in Section 4.2 are not required to apply our framework – we select a task where this division is clear for our experiments only because it better motivates our study of task accuracy.
> - Our framework *does* make one important assumption regarding properties: that is, we assume that there is a finite set of properties (i.e., discrete random variables $\mathbf{Z} = Z_1, …, Z_k$ corresponding to latent features of text inputs) that each take a specific value $Z_i = z$ for any given input $x$. However, this is simply the definition of probing (Belinkov, 2022), meaning that all works in the probing literature share this assumption. As such, the framework can be applied without modification to any setting that can be studied via causal probing (as discussed in Appendix A).
>
> However, we agree that the specific experimental instantiation of our framework, as explored in this work, is somewhat limited in that we only experiment with a single well-studied language model, a simple linguistic task, and two binary variables denoting surface-level syntactic properties. We begin with this simple setting because, despite the abundance of interpretability research on this model and task, there is still no consensus regarding what causal probing methods are appropriate for studying this setting, let alone how to compare and evaluate such methods. Thus, we argue that it is important to begin by comprehensively studying existing causal probing methods using the proposed framework in this simple setting, rather than prematurely scaling our analysis to more complex models and tasks (see Appendix A). Note that this choice also enables us to study existing methods in contexts for which they were originally designed, rather than applying them in new scenarios for which they were never intended (as doing so might have yielded unfair results).
>
> **C2: Challenging in practice: approximating oracle probes**
>
> You note that “some ideas seem challenging to achieve in practice (line 145)”, and the challenge described in this part of the paper is that “a perfect oracle probe does not exist in practice, so any practical implementation must approximate it.”
>
> It is true that our framework crucially relies on high-quality oracle probe approximations, and the question of how best to evaluate such approximations is indeed an open one. In our experiments, we simply perform standard hyperparameter grid search over possible oracle probes and select the MLP architecture and learning rate that yields the highest validation-set accuracy (see Section 4.3 and Appendix C.2). We opted for this approach in accordance with Occam’s razor: as we are the first to define and experiment with oracle probes, we selected from the range of possible oracle probes in the way that one would select any other model or probe (i.e., the one with the best performance on the validation set).
>
> However, we also experimented with linear oracle probes (see Appendix D.3). These linear oracles yielded similar results to those obtained using the MLP oracles whose results are listed in the main paper – e.g., all major findings are the same – which gives us some confidence that different types of oracle probe approximations generally yield similar results in the experimental setting explored in this work. However, we acknowledge that this is a purely empirical observation made on the basis of a limited set of experiments, and it is possible that the matter of oracle probe selection could be much more consequential in more complex experimental scenarios. This is an important consideration for future work, and we will add further discussion on this point to the paper.
>
> **C3: Redundant descriptions**
>
> Thank you for pointing out that there is some overlapping and redundant information in Sections 4 and 5, and that Section 5 contains some excessive experimental details. We agree, and will remove redundancies and port some minor details to the appendix.
>
> (Continued in rebuttal post 2/N.)

---

> ### Author Response · Authors · 2024-11-21
> **Rebuttal (part 2/N)**
>
> (Continued from rebuttal post 1/N.)
>
> **C4: Experiments assume full determinism and binary variables**
>
> If, by “full determinism”, you mean access to ground-truth labels for properties, where properties are discrete random variables taking a specific (deterministic) value for any given token or token sequence, then it is true that this is an assumption of our framework, but this assumption is common to all probing research, as discussed in our response to **C1**. (Please let us know if you meant something else by “full determinism”, and we will be happy to discuss this matter further.)
>
> Next, while it is true that our *experiments* include only binary variables, our *framework* is explicitly defined to accommodate properties with more than two possible variables – see lines 178-180.
>
> Finally, our framework is also general to any number of properties – see, e.g., line 200-203, which defines selectivity when more than two properties are being considered (the definition for completeness and reliability does not change). Thus, while our experiments only consider a scenario with two properties, $Z_c$ and $Z_e$ in the context of the subject-verb agreement task, our framework can still be applied to any other language-modeling task, and also does not require that properties be categorized as “causal” or “spurious” (as noted in our response to the first concern).
>
> We will update Sections 3 and 4 to more explicitly differentiate between the “probing assumption” required by our *framework* versus the features of our *experimental setting* that are not assumed by the framework.
>
> **Conclusion**
>
> We hope we have resolved all of your concerns. If not, we remain available for further discussion.

---

> > ### Author Response · Authors · 2024-11-27
> > **Request response to our rebuttal**
> >
> > Once again, we thank you for your initial review of our paper. To promote a productive discussion period, we would greatly appreciate it if you are able to respond to our rebuttal.
> >
> > Have we fully resolved all of your concerns, or do you have any additional questions? We are more than happy to clarify any potential points of confusion and engage in further discussion.

---

> ### Comment · Reviewer_vDrB · 2024-12-03
>
> Thank you for your response. However, considering the gap between the proposed framework and the experimental setting, I will keep my socre.

---

> ### Author Response · Authors · 2024-12-04
> **Please take rebuttal into consideration during meta-review**
>
> Dear Reviewer,
>
> Thank you for responding to our rebuttal. We respectfully disagree with your characterization of our experimental setting: it is a valid instantiation of the proposed framework, with no "gap" between the framework and experiments.
>
> Rather, as you expressed in your initial review, the potential concern here is whether the experiments are sufficiently broad to validate our general framework. We responded to this point in detail in our rebuttal (see **C1, C2,** and **C4**). We regret that there was not time to engage in further discussion on this matter, given that the response to our rebuttal was posted with less than 3 hours left in the discussion period and does not contain any specific feedback, suggestion, or followup question for us to respond to on the final day for author responses.
>
> Although we were unable to engage with you during the discussion period, we kindly request that you please consider whether our rebuttal addressed the important potential concerns you posed in your initial review, and hope that you will update your assessment during the meta-review process accordingly.
>
> Thank you,
>
> Authors

---

### Official Review · Reviewer_pbgx · 2024-11-08

**Soundness:** 1
**Presentation:** 3
**Contribution:** 1
**Rating:** 3
**Confidence:** 3

**Summary:**

This paper proposes a method to evaluate interventions in a model’s representation to enforce specific properties. The authors propose to evaluate these interventions by training a model evaluator (approximated oracle probe) that can assess whether the targeted property has been set or removed, as well as the degree to which the intervention affects other properties. The evaluation is conducted on a single-task case study where the target property has two possible values.

**Strengths:**

The paper is well-written and organized, making the proposed method easy to follow.

**Weaknesses:**

The primary concern with this work lies in the overlap between the evaluator (approximated oracle probe) and the intervention generator (underlying structural probe used, for example, in the counterfactual approach). Both aim to predict the latent variable \( Z \) from the representation, meaning the evaluator and the intervention generator essentially encode the same information. This overlap likely explains why the counterfactual intervention seems to perform best, as assuming they both perform well (also, the authors use the same architecture for these models), it is almost like directly leveraging the evaluator in generating the intervention and assessing using the evaluator afterwards.

The simplicity of the experimental setup also limits the work's scope. It would be more compelling if the authors tested their method on cases with more than binary causal variables. Additionally, introducing interventions that could potentially not impact task accuracy would provide a clearer understanding of whether performance loss is due to the intervention’s effectiveness or simply the disruption caused by perturbing the representation.

Another suggestion would be to evaluate a scenario where \( Z_c \) and \( Z_e \) are independent, \( Z_c \) only impact label. This setup could test if intervening on \( Z_e \) affects task accuracy while maintaining good selectivity or completeness.

**Questions:**

- The rationale behind equating nullified representations to a uniform distribution isn’t entirely clear (I have already read your detailed description in Appendix B). Why not add a distinct category representing "neither" of the options, rather than assuming a uniform distribution?

---

> ### Author Response · Authors · 2024-11-21
> **Rebuttal (part 1/N)**
>
> Thank you for your thoughtful review. We appreciate that you found our paper to be “well-written and organized,” and our method to be “easy to follow.”
>
> We address each of the concerns you raised as follows:
>
> **C1: Relationship between oracle probes and interventional probes**
>
> It is true that the “evaluator” (approximated oracle probe) and “intervention generator” (probes used to obtain interventions) are trained on very similar tasks: in each case, to predict the value of a property $Z_i$ (i.e., a discrete random variable denoting an input feature) from language model embeddings. This is intended: oracle probes are used to validate whether or not an intervention has had the desired effect on the representation of the target property encoded by embeddings. There are several important details concerning the relationship between oracles and interventional probes that make oracle probes appropriate for evaluating interventions:
>
> 1. As noted in lines 282-285, there is no overlap between the data used to train or validate approximated oracle probes versus interventional probes, and the test data used to evaluate interventions is also fully disjoint from the training and validation set of both kinds of probes.
> 2. Oracle probes used to measure *completeness* are indeed trained to predict the same property $Z_i$ as interventional probes. However, at test time, interventional probes are used to modify embeddings to encode a different value of $Z_i$ ($z' \neq z$ for counterfactual interventions, and $z’=0$ for nullifying interventions; see Section 3), and oracle probes are used to validate the extent to which the intervention has carried out the desired operation. Thus, it is in no sense “guaranteed” that interventions performed using a given interventional probe will produce embeddings that are recognized by an oracle probe as encoding the target value (i.e., completeness = 1); and as observed in our experiments, oracle probes usually do *not* show that the desired operation has been fully carried out.
> 3. Oracle probes used to measure *selectivity* are trained to predict a *different* property $Z_j$, while interventional probes are trained to predict property $Z_i$, and $Z_j \neq Z_i$. This means that there is no “overlap” between the tasks learned by oracle probes measuring selectivity and interventional probes. Thus, given a perfect intervention method, interventions performed using this method over a given interventional probe (trained to predict $Z_i$) should not affect the oracle probe’s prediction of $Z_j$.
> 4. As explained in lines 285-290, there is an important difference between the training distributions of interventional and oracle probes:
>     - In order to study interventions in a realistic setting (where it is generally not possible to control for all non-targeted properties), *interventional probes are trained with data that is fully i.i.d. with respect to the data distribution*, meaning that they might incorrectly leverage spurious correlations with $Z_j$ to predict the target property.
>     - On the other hand, *oracle probes are trained on data that is subsampled in order to enforce conditional independence* between a target property $Z_i$ and the other properties $Z_j \neq Z_i$ that are considered in measuring selectivity. This means that oracle probes are unable to leverage spurious correlations between these properties, which is important for ensuring that oracle probes can serve as objective evaluators of interventions.
>
> (Continued in rebuttal post 2/N.)

---

> ### Author Response · Authors · 2024-11-21
> **Rebuttal (part 2/N)**
>
> (Continued from rebuttal post 1/N.)
>
> **C2: Are oracle probes biased in favor of counterfactual interventions over nullifying interventions?**
>
> First, we will describe our understanding of your specific concern as follows – please let us know whether or not our understanding is correct:
> - As discussed in Section 3, both (a) oracle probe training distributions $P_o(Z | h)$ (where $h$ is an embedding), and (b) target counterfactual distributions (where probability mass is shifted from the original ground-truth value of a property to a counterfactual value), are “one-hot” distributions (where probability is 1 for a given value and 0 for all others), whereas nullifying interventions are uniform.
> - Thus, because the training objective of oracle probes has the “same shape” (one-hot) as counterfactual distributions, this might be interpreted as biasing oracle probes in favor of counterfactual interventions over nullifying interventions.
>
> However, there are two reasons that this interpretation is incorrect:
> 1. Nullifying interventions are obtained using interventional probes that are also trained on “one-hot” distributions, and these interventions explicitly operate by modifying embeddings such that these interventional probes predict the uniform distribution over possible values. For instance, INLP projects embeddings into the nullspace of linear interventional probes, guaranteeing that each of the interventional probes will assign equal probability to each possible value of the target property. Thus, it is entirely appropriate to evaluate such interventions on the basis of how similar an impact they yield on oracle probes, in precisely the same way that counterfactual interventions are evaluated based on whether they have the same impact on oracle probes as the interventional probes used to perform them.
> 2. One possible response to (1) is that, e.g., INLP is only designed to induce a uniform distribution over *linear* interventional probes, whereas the oracle probes used in the experiments reported in the main paper are MLPs. However, AlterRep also only considers linear interventional probes – in fact, in our implementation, they are the *exact same probes* – and while AlterRep yields the greatest “peak” reliability of all interventions (using an MLP oracle probe), INLP performs worse than all counterfactual methods (see Table 1). Furthermore, in Appendix D.3, we also perform the same experiments using linear oracle probes, and observe similar results: the same tradeoff exists between completeness and selectivity, and comparing intervention methods by maximum reliability yields the same ordering of most- to least-reliable (in particular, still placing both nullifying methods below all counterfactual methods). Therefore, our finding that nullifying methods are less complete and reliable than counterfactual methods is independent of whether the approximated oracle probe is linear or nonlinear.
>
> **C3: Simplicity of experimental setup (binary variables)**
>
> First, it is important to clarify that our *experiments* are restricted to binary random variables, but our *framework* is explicitly defined to accommodate variables with more than two possible variables – see lines 178-180.
>
> However, we agree that the specific experimental instantiation of our framework, as explored in this work, is somewhat limited in that we only experiment with a single well-studied language model, a simple linguistic task, and two binary variables denoting surface-level syntactic properties. We begin with this simple setting because, despite the abundance of interpretability research on this model and task, there is still no consensus regarding what causal probing methods are appropriate for studying this setting, let alone how to compare and evaluate such methods. Thus, we argue that it is important to begin by comprehensively studying existing causal probing methods using the proposed framework in this simple and well-studied setting, rather than prematurely scaling our analysis to more complex models and tasks (see Appendix A). Note that this choice also enables us to study existing methods in contexts for which they were originally designed, rather than applying them in new scenarios for which they were never intended (as doing so might have yielded unfair results).
>
> (Continued in rebuttal post 3/N.)

---

> ### Author Response · Authors · 2024-11-21
> **Rebuttal (part 3/N)**
>
> (Continued from rebuttal post 2/N.)
>
> **C4: Introducing task-irrelevant interventions**
>
> In the review, you suggested we introduce “interventions that could potentially not impact [language model] task accuracy” in order to “provide a clearer understanding of whether performance loss is due to the intervention’s effectiveness or simply the disruption caused by perturbing the representation.” However, it is not entirely clear to us what such an intervention would be, as any intervention over embedding representations requires perturbing them, and will correspondingly lead to performance loss if the perturbation magnitude is too great.
>
> Our best interpretation is that we might consider interventions that are completely unrelated to task-relevant properties. As a random baseline to better understand an appropriate range for perturbation magnitude $\epsilon$ used to constrain GBIs, we have run an additional experiment with interventions that randomly perturb each element of an embedding vector by $+\epsilon$ or $-\epsilon$. (Note that this is similar to FGSM GBIs, which also yield perturbation vectors where each element is positive or negative $\epsilon$, but here positive and negative values are randomly determined instead of setting them to the sign that most increases probe loss.)
>
> The results are visible at [this anonymized link](https://osf.io/7xj6w?view_only=b9189a028ce5434cbf4ece37ca8e651b), where we plot the reliability (solid line) versus $\Delta$Acc of this random perturbation by $\epsilon$ value in purple, and include FGSM for comparison. (For reference, this plot corresponds to Figure 6 in the appendix.)
>
> In these results, we observe that the random intervention has both a much lower reliability, and a lower impact on LM task accuracy, relative to FGSM for a given $\epsilon$. As such, this may be taken as evidence that LM performance loss is indeed primarily due to the specific representation targeted by FGSM, and not simply the disruption caused by perturbing the embedding.
>
> **C5: Considering scenarios where properties are independent**
>
> You recommended that we “evaluate a scenario where $Z_c$ and $Z_e$ are independent, $Z_c$ only impact label.” As explained in Section 4.2, this is precisely the scenario we consider in all experiments: by definition, $Z_c$ causally determines the ground-truth label in the subject-verb agreement task, and $Z_e$ is spurious (meaning that it may be correlated with the label, but does not determine it).
>
> **C6: Evaluate completeness of nullifying methods by adding a new class for oracle probes?**
>
> We evaluate nullifying interventions by comparing the probability distribution emitted by an oracle probe (using the nullified embedding as input) to the uniform distribution. You suggested that instead we might “add a distinct category representing ‘neither’ of the options, rather than assuming a uniform distribution”; but this would be ill-defined for two-reasons:
>
> 1. As discussed in our response to **C2**, nullifying interventions like INLP work by modifying embeddings such that interventional probes predict the uniform distribution over possible values, meaning that it is appropriate to evaluate such interventions on the basis of how similar an impact they have on oracle probes as they (are designed to) have on interventional probes.
> 2. It would also be impossible to train probes on the proposed “none of the above” category. Probing is defined as learning a mapping from an embedding $h$ to a property (discrete random variable) $Z_i$ that takes a specific value $Z_i = z$ for any given input $x$ (Belinkov, 2022). Thus, in order to train an oracle probe with respect to the “none of the above” class, it would be necessary to provide models with an input $x_0$ where $Z_i$ does not take any value; but by definition, there exists no such $x_0$ in English. (See lines 484-488 for further discussion.)
>
> **Conclusion**
>
> We hope we have resolved all of your concerns. If not, we remain available for further discussion.

---

> > ### Comment · Reviewer_pbgx · 2024-11-25
> > **Relationship between oracle probes and interventional probes is still not clear.**
> >
> > Thank you for your answers!
> >
> > I am going to focus on the most critical issue: the relationship between oracle probes and interventional probes.
> >
> > Let’s take the example of oracle probes used for completeness. If I understand correctly, your main argument is that the data used are different, so they may learn different functions. However, the end task is exactly the same. If we set aside the data for a moment and take your example, assuming they have perfectly learned from their respective data. Both models are essentially approximating the same ground truth function. My point remains that using oracle probes is almost equivalent to using the intervention model as both the evaluator and for the interventions.
> >
> > On this note, the second approach that enforces conditional independence also seems a good method for learning the intervention model. Why didn’t you use it to learn the intervention probes? Why allow the intervention probes to learn spurious correlations and not the oracle probes, when at the end, they are all trying to predict the same thing?
> >
> > Regarding the statement: “Thus, it is in no sense ‘guaranteed’ that interventions performed using a given interventional probe will produce embeddings that are recognized by an oracle probe as encoding the target value (i.e., completeness = 1).”
> >
> > I agree with this point. However, my argument is that you are using the same model to learn the same task (albeit with different data). That doesn’t change the fact that they are both trying to approximate the same function. It makes sense to expect that they can converge to the same model, and therefore, it is likely that the interventions extracted by the intervention model might work for your oracle probes.

---

> > > ### Author Response · Authors · 2024-11-27
> > > **Response to "Relationship between oracle probes and interventional probes is still not clear."**
> > >
> > > Thank you for your response to our rebuttal! We see two remaining concerns in in your response, and address them individually below:
> > >
> > > **C1: Oracle probes and interventional probes learn the same task.**
> > >
> > > We agree with your statement that if we “set aside [training] data,” then “using oracle probes is almost equivalent to using the intervention model as both the evaluator and for the interventions.” As noted in our rebuttal, this is intended: the purpose of oracle probes is to test whether an intervention that “fools” an interventional probe (causing it to predict the target counterfactual or nullified value) also “fools” the approximated oracle probe, given that both kinds of probes have indeed been trained to perform the same task. If the oracle probe is perfectly “fooled”, then the completeness score of the intervention is 1.
> > >
> > > However, there are a variety of potential limitations of causal probing interventions that might result in the oracle probe not being “fooled”. For instance:
> > > - Interventional probes might overfit their training data or rely on spurious information that is not learned by oracle probes (see our response to **C2**).
> > > - Interventions might not fully transform the representation of the target property learned by the interventional probe.
> > > - Interventional probes might be subject to different adversarial vulnerabilities (as exploited by, e.g., GBIs, which use adversarial attacks against interventional probes) than oracle probes.
> > >
> > > In each of these cases, the intervention would be incomplete, and an ideal oracle probe would show this. Indeed, for most interventions and hyperparameter settings, we observe this result – i.e., it is uncommon for approximated oracle probes to show that an intervention is complete or nearly-complete; and in all such cases, we see a corresponding drop in selectivity. This indicates that, despite learning the same task as interventional probes, our approximated oracle probes are nonetheless capable of revealing limitations of current intervention methods.
> > >
> > > **C2: Why not enforce conditional independence over training data of interventional probes?**
> > >
> > > Our goal is to study causal probing methods in a realistic setting where probe training data is not carefully controlled with respect to all potential spurious correlations. It is not feasible to control for all such correlations in practice — e.g., there are an unbounded number of properties that *might* be relevant to any given task, annotating probe training data for each property would be an extensive annotation effort, etc.; and we are not aware of any prior work in causal probing that has controlled for spurious correlations in probe training data in this way. As such, we may summarize a practical assumption common to prior work as follows: *the presence of spurious correlations in their probe training data does not affect their main findings.* (E.g., this would be true of perfectly selective methodologies.)
> > >
> > > How valid is this assumption in practice? To answer this question, we enforce conditional independence over oracle probe training data while *not* doing so for interventional probes, which allows us to use oracle probes to study how spurious correlations in interventional probe training data impact the selectivity of downstream interventions. We find that the answer varies widely by method: e.g., INLP, which explicitly minimizes “collateral damage” to embeddings, retains high selectivity despite spurious correlations in interventional probes’ training data; but for GBIs, which are not optimized in this way, we see a large drop in selectivity when hyperparameters are calibrated for maximum reliability or completeness (see Table 1).
> > >
> > > **Conclusion**
> > >
> > > Please let us know if we have fully addressed your concerns, or if you have any remaining questions. We remain available for further discussion.

---

> > > > ### Author Response · Authors · 2024-12-04
> > > >
> > > > Dear Reviewer,
> > > >
> > > > Once again, thank you for responding to our rebuttal. We hope that our last reply fully addressed your remaining concerns. If so, we would like to kindly request that you please update your assessment accordingly during the meta-review process.
> > > >
> > > > Thank you,
> > > >
> > > > Authors

---

### Official Review · Reviewer_hJpz · 2024-11-15

**Soundness:** 2
**Presentation:** 4
**Contribution:** 3
**Rating:** 5
**Confidence:** 2

**Summary:**

The paper studies about existing causal probing methods through the lens of completeness and selectivity. Here, completeness measures how effectively the model’s representations capture the target property and selectivity measures the undesirable impact of the intervention on irrelevant property. The implication is that there exist certain tradeoff between completeness and selectivity and the authors further suggest that causal probing approaches with counterfactual interventions are more effective compared to approaches with nullifying interventions.

**Strengths:**

- The paper studies important and timely topic, i.e., the thorough examination of existing causal probing methods, in a principled way using two measures (completeness and selectivity). These metrics seem to be intuitive and reasonable.
- The paper is well-written and well-structured. The paper is easy to follow and the authors provide detailed background which makes the reader outside of the field easy to understand the problem and their analysis.

**Weaknesses:**

- The paper does not provide actionable solution for improving existing approaches. The implication of the study in this paper suggests that the causal probing approaches with counterfactual interventions might be more effective compared to approaches based on nullifying interventions, but I believe the paper would further strengthened if the authors could discuss the possible directions for (potentially) improving existing probing methods.
- The framework heavily relies on the approximated oracle probe, i.e., the most expressive probe, but it is unclear whether such assumption holds or not. Specifically, the authors acknowledge that there should be no spurious correlation between $Z_c$ and $Z_e$ for training the oracle probe, but it is unclear how to measure such correlation and whether this approach is always feasible in more realistic scenarios. Finally, some theoretical guarantee on the approximated oracle would further strengthen the paper (e.g., clearly stating the assumptions, discussion on the assumption violations, estimator error, etc)

**Questions:**

- See above

---

> ### Author Response · Authors · 2024-11-21
> **Rebuttal**
>
> Thank you for your thoughtful review. We appreciate that you find the topic of our paper to be “important and timely,” that our proposed metrics are “intuitive and reasonable,” and that our paper is “well-written and well-structured” and “easy to follow.”
>
> We address each of the concerns you raised as follows:
>
> **C1: Can the authors discuss possible directions for improving existing approaches?**
>
> While the primary goal of our work is to define an empirical analysis framework to measure the reliability of causal probing methods, and not explicitly to improve the methods under investigation, we do see several ways our framework could be leveraged to improve causal probing methodologies:
>
> 1. *Our framework allows one to calibrate the hyperparameters of existing causal probing interventions to optimize completeness, selectivity, or the compromise between them (reliability).* Previously, it has not been clear how to define, measure, or balance such priorities, and prior work has resorted to a variety of differing heuristic methods for determining such hyperparameters (see, e.g., Ravfogel et al., 2020, 2021; Elazar et al., 2021; Tucker et al., 2021; Davies et al., 2023, etc.). In our experiments, we show that there is a tradeoff between completeness and selectivity; and we compare methods by setting their hyperparameters to the values that yield the maximum reliability, demonstrating that our framework can be leveraged as the first concrete basis for optimizing intervention hyperparameters.
>
> 2. *How can we objectively measure and compare the “success” of causal probing methods, and how can we make progress as a community if we are unable to do so?* It has previously been unclear how to quantify or demonstrate improvements offered by new methods, given the lack of any clear metric for comparing them or measuring progress. Our framework makes it possible, for the first time, to directly compare the reliability of various causal probing methods, providing future work with a clear basis to show whether or not a new approach improves over existing methods.
>
> We will add discussion on both of these points to our conclusion..
>
> **C2: Reliance on approximated oracle probe**
>
> It is true that our framework crucially relies on oracle probe approximations, and the question of how best to obtain and evaluate such approximations is indeed an open one. In our experiments, we simply perform standard hyperparameter grid search over possible oracle probes and select the architecture and learning rate that yields the highest validation-set accuracy (see Section 4.3, Appendix C.2, and Appendix D.3). We opted for this approach in accordance with Occam’s razor: as we are the first to define and experiment with oracle probes, we selected from the range of possible oracle probes in the same way that one would typically select any other model or probe – i.e., the one with the best performance on the validation set. However, we did also carry out experiments with some of the less-performant probe architectures (see Appendix D.3), yielding similar results to those obtained using the top-performing oracles whose results are listed in the main paper.
>
> **C3: Oracle probe assumptions**
>
> We enumerate the full set of *experimental assumptions* for oracle probe approximations in Section 4.3 (lines 282-290) – i.e., that they should be trained on a disjoint set from interventional probes, that the target property should be conditionally independent of the other properties, and that the label distribution of the target property should be preserved while enforcing conditional independence. However, please note that these are not *theoretical assumptions* for defining oracle probes – rather, they are empirical measures to obtain more reliable oracle probe approximations (see lines 285-290).
>
> We will better clarify this point in Section 4.3 – specifically, that these experimental assumptions are introduced as initial “best practices”, and that future work could experiment with additional measures to improve oracle probe approximations.
>
> **Conclusion**
>
> We hope we have resolved all of your concerns. If not, we remain available for further discussion.

---

> > ### Author Response · Authors · 2024-11-27
> > **Request response to our rebuttal**
> >
> > Once again, we thank you for your initial review of our paper. To promote a productive discussion period, we would greatly appreciate it if you are able to respond to our rebuttal.
> >
> > Have we fully resolved all of your concerns, or do you have any additional questions? We are more than happy to clarify any potential points of confusion and engage in further discussion.

---

> > > ### Author Response · Authors · 2024-12-04
> > > **Please take rebuttal into consideration during meta-review**
> > >
> > > Dear Reviewer,
> > >
> > > We appreciate your initial review of our paper, and regret that we did not receive a response to our rebuttal (posted Nov 20). We hope that our rebuttal fully addressed your concerns.
> > >
> > > Although we were unable to engage with you during the discussion period, we would like to kindly request that you please update your assessment during the meta-review process based on the content of our rebuttal. If we were able to resolve your initial concerns, we hope that you will take this into account.
> > >
> > > Thank you,
> > >
> > > Authors

---

### Author Response · Authors · 2024-12-02
**Final day of discussion period: request responses to rebuttal**

Dear AC and Reviewers,

We posted a rebuttal to each of the 5 initial reviews 12 days ago (Nov 20), but we still have not received a response from 3/5 reviewers. *As the discussion period ends today (AoE),* **we request that the remaining three reviewers please respond to our rebuttal** and let us know if we have fully resolved their initial concerns, or if they have any further questions or feedback. As there is an additional 24 hours for *authors*  to respond to final discussion posts from reviewers, *it is essential that **reviewers** respond today* so that we will be able to provide any final answers or clarifications tomorrow.

Additionally, to facilitate a fair and productive review process, **we would like to kindly request that the area chair please invite these reviewers to engage with our rebuttal before the discussion period ends**.

Finally, for the two reviewers who have responded to our rebuttal: thank you for your initial response! *Have we fully resolved the followup questions you raised, or do you have any remaining concerns?*

Thank you very much for your time and attention on this matter.

Best regards,

Authors

---

### Meta-Review · Area_Chair_Anfa · 2024-12-21

**Metareview:**

The paper introduces oracle probes to evaluate causal probing methods by assessing completeness (capturing targeted properties) and selectivity (avoiding impact on irrelevant properties).


Strengths:

+ Provides an evaluation framework for causal probing methods that could standardize comparisons across approaches.

Weaknesses:

+ Relies on assumptions about the validity of oracle probes without theoretical or robust empirical validation (e.g., handling spurious correlations).

+ Experiments are limited in scope, focusing on binary causal variables and simple tasks, which limits generalizability.

+ Does not offer actionable solutions for improving existing probing methods or scaling the framework to more complex scenarios.

**Additional Comments On Reviewer Discussion:**

The reviewers had a consensus on the potential of the proposed method but hesitated to recommend acceptance without further validation or theoretical grounding. They remain concerned about the heavy reliance on oracle probes, yet no theoretical validation or robust empirical evidence of their validity. Empirical results are also limited by using validation-set accuracy, which cannot rule out the influence of spurious correlations or confirm generalization to out-of-distribution scenarios.

---

### Decision · Program_Chairs · 2025-01-22

Reject